# EXPLORING SOLUTION DIVERGENCE AND ITS EFFECT ON LARGE LANGUAGE MODEL PROBLEM SOLVING

## ABSTRACT

Large language models (LLMs) have been widely used for problem-solving tasks. Most recent work improves their performance through supervised fine-tuning (SFT) with labeled data or reinforcement learning (RL) from task feedback. In this paper, we study a new perspective: the divergence in solutions generated by LLMs for a single problem. We show that higher solution divergence is positively related to better problem-solving abilities across various models. Based on this finding, we propose solution divergence as a novel metric that can support both SFT and RL strategies. We test this idea on three representative problem domains and find that using solution divergence consistently improves success rates. These results suggest that solution divergence is a simple but effective tool for advancing LLM training and evaluation.

## 1 INTRODUCTION

The rise of large language models (LLMs) and their remarkable general problem-solving capabilities have accelerated research on advanced artificial intelligence (AI) solutions across diverse domains, including science (Ren et al., 2025), finance (Li et al., 2023), and education (Wang et al., 2024). In particular, problems in STEM subjects such as mathematics (Liu et al., 2024), logic reasoning (Parmar et al., 2024) and programming (Coignion et al., 2024) have received significant attention, as their solutions can be objectively verified. A wide range of advanced algorithms have been proposed to improve LLMs' problem-solving success, most of which focus on either expanding training datasets or applying supervised fine-tuning (SFT) with step-by-step annotated solutions (Zhang et al., 2024) or employing reinforcement learning (RL) with correctness-based rewards (Ouyang et al., 2022).

While these approaches highlight the value of data in improving LLM performance, in this work we turn to an underexplored property shared across problem-solving datasets: the solution divergence, which refers to the presence of multiple viable solutions to a single problem. Studying solution divergence offers two key benefits in improving the models' performance. First, the majority of existing work in using SFT to boost LLM's performance on the task relies on generating or collecting new problems to augment training data, a process that is costly and labor-intensive due to the need for extensive cleaning and quality control (Shen, 2024). Although synthetic approaches such as question paraphrasing have been proposed (Chen & Lin, 2024), they often produce inconsistent quality and risk diverging from authentic problem distributions, limiting their effectiveness (Chen et al., 2024b). By contrast, leveraging solution divergence allows us to enrich datasets using existing, authentic problems, thus avoiding these drawbacks. Second, it is evident from cognitive science research that humans with larger repertoires of problem-solving strategies perform more effectively on complex tasks (Siegler, 1998). Given the growing similarities between human and model problem-solving behaviors, such as step-by-step reasoning (Wei et al., 2022), we argue that studying the solution divergence potentially offers a new perspective for understanding LLM behavior from a cognitive-science point of view. Moreover, it provides new opportunities to improve LLM problem-solving performance. For instance, education research shows that fostering solution diversity in learners leads to better academic outcomes (Caviola et al., 2018). By integrating solution divergence, we can analogously enhance LLMs.

Overall, our study is organized as follows. In Section 3, we define the concept of solution divergence and evaluate it on three representative problem-solving datasets spanning mathematics, programming, and logical reasoning, examining its relationship with LLM performance across mul-

tiple models. In Section 4, we propose methods to incorporate solution divergence into both SFT-and RL-based fine-tuning paradigms. Finally, in Section 5, we present experiments on datasets from different domains and empirically demonstrate that integrating solution divergence enhances the problem-solving capabilities of LLMs.

## 2 RELATED WORK

**LLM Optimization.** LLM optimization methods typically build on SFT and RL. SFT has proven effective across domains such as coding (Roziere et al., 2023), mathematics Hendrycks et al. (2021); Toshniwal et al. (2024), and general reasoning (Yue et al., 2024), where curated datasets like Code Llama, OpenMathInstruct-2, and MAmmoTH2 yield substantial improvements. Reinforcement learning from human feedback (RLHF) (Ouyang et al., 2022) further established a widely used paradigm for aligning models to human preferences. More recently, group-based RL variants have been proposed to better capture sequence-level reasoning: GRPO (Shao et al., 2024) introduces group-wise optimization to improve mathematical reasoning, DAPO (Yu et al., 2025) refines stability for long chain-of-thought training via dynamic sampling and token-level gradients, and GSPO (Zheng et al., 2025) adopts sequence-level clipping for greater efficiency. Our work complements these approaches by introducing solution divergence as an explicit signal, used both for selecting training samples in SFT and for designing diversity-aware reward functions in RL.

**Data Diversity in LLM Training and Inference.** Parallel lines of research emphasize the role of data and output diversity. At the training stage, diverse prompts and responses have been shown to enhance robustness and alignment (Bukharin & Zhao, 2023; Song et al., 2024), while synthetic data studies also report strong links between diversity and downstream generalization (Chen et al., 2024a). At inference, prompt ensembles, sampling strategies, and temperature scaling are commonly used to elicit multiple solution paths (Kirk et al., 2023). Unlike these heuristic approaches, our framework formalizes diversity through a measurable solution divergence metric and integrates it directly into training objectives, unifying dataset-level diversity with inference-time diversity in a principled manner.

## 3 PRELIMINARY STUDY

### 3.1 SOLUTION DIVERGENCE DEFINITION

We consider a question dataset $\mathcal{Q} = \{q_n \mid n = 1, \ldots, N\}$ of size N. For each question $q_n$, the LLM ($\pi_\theta$) generates a solution set $\mathcal{S}_{q_n} = \{s_m \mid m = 1, \ldots, M\}$ of size M. The divergence between two solutions $s_i$ and $s_j$ is represented by $\delta_{i,j}$, capturing the degree of difference between them. For each question $q_n$, the overall divergence of its solution set $\mathcal{S}_{q_n}$ is denoted as $\zeta_{q_n}$. The model's solution divergence over $\mathcal{Q}$ is written as $\zeta_\pi = \text{mean}(\zeta_{q_n})$. In cognitive science studies (Caviola et al., 2018), the pairwise divergence $\delta_{i,j}$ is represented as a binary value $\{0, 1\}$, where 0 indicates identical solutions and 1 indicates different solutions. These judgments are usually made by experts through manual review of solution pairs. Based on this, $\zeta_{q_n}$ is calculated by counting the number of unique solutions in the set of solutions. Formally, this can be expressed by constructing a weighted relation graph $\mathcal{G}$, where nodes correspond to solutions and edge weights correspond to their similarity, i.e., $1 - \delta_{i,j}$ for $s_i$ and $s_j$. The number of connected components in $\mathcal{G}$ then yields $\zeta_{q_n}$.

However, in our study, the scale of LLM-generated solutions makes manual labeling infeasible. To address this, we proxy $\delta_{i,j}$ by the normalized string edit distance $d^{(e)}$:

$$\delta(s_i, s_j) = \frac{d^{(e)}(s_i, s_j)}{\max(|s_i|, |s_j|)} \tag{1}$$

where $|\cdot|$ denotes text length. Intuitively, the more overlapping characters two solutions share, the smaller their divergence. We note that string edit distance provides only a restricted perspective on divergence due to the inherent flexibility of natural language. Nevertheless, we adopt it because of its computational efficiency and consistency across domains. This is important since

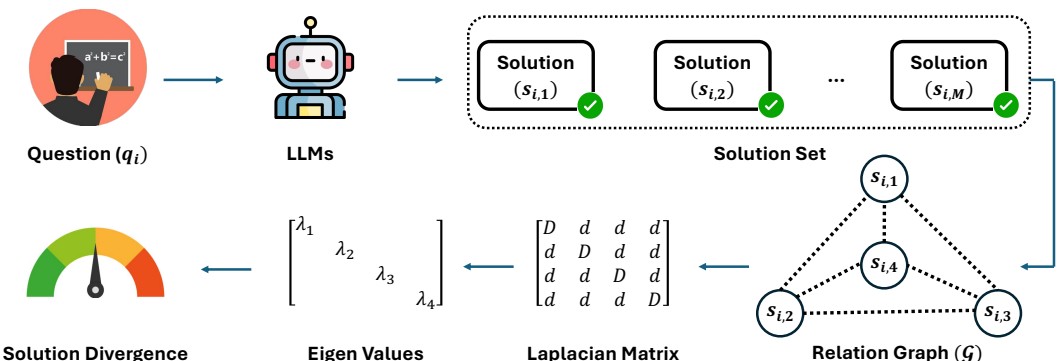

Figure 1: An Overview of the Solution Divergence Calculation

solution formats can differ greatly (e.g., mathematical derivations versus programming code), making domain-specific divergence metrics inefficient to deploy adaptively. Thus, $\delta_{i,j}$ offers a practical means to analyze the relationship between solution divergence and problem-solving performance. More advanced proxy metrics will be explored as one future work.

To calculate $\zeta_{q_n}$, we derived it from the eigenvalues $\Lambda = \{\lambda_1, \ldots, \lambda_M\}$ of the Laplacian matrix $L$ of the relation graph $\mathcal{G}$. This adjustment is necessary because the edge weights $\delta_{i,j}$ are non-binary. Inspired by spectral clustering (Von Luxburg, 2007), where the magnitude of eigenvalues reflects the tightness of clusters in a relational graph, we propose two variants:

$$\zeta_{q_n}^l = M - \lambda_2, \quad \zeta_{q_n}^g = M - \frac{1}{M} \sum_{i=1}^{M} \lambda_i. \tag{2}$$

The local variant $\zeta_{q_n}^l$ is highly sensitive to weak local connections; even small changes in the smallest $\delta_{i,j}$ can substantially affect its value. By contrast, the global variant $\zeta_{q_n}^g$ captures overall graph tightness from a global perspective. The solution divergence of an LLM is denoted as $\zeta_\pi^l$ and $\zeta_\pi^g$, respectively. A schematic illustration of the solution divergence calculation is shown in Figure 1.

## 3.2 STUDY SETTINGS

**Datasets**   To comprehensively study the relationship between solution divergence and LLM performance, we employ three representative problem-solving datasets: Math-500 (Lightman et al., 2023), MBPP+ (Liu et al., 2023), and Maze. Math-500 and MBPP+ are well-established benchmarks for evaluating LLMs in mathematical problem solving and automatic code generation, respectively. In addition, we introduce Maze, a novel logical reasoning dataset developed for this study. Each problem in Maze requires the model to identify a viable path from a given start point to an endpoint on a 2D coordinate grid while avoiding

Table 1: Example questions from the datasets.

| Dataset | Question |
|---|---|
| Math-500 | A regular hexagon can be divided into six equilateral triangles. If the perimeter of one of the triangles is 21 inches, what is the perimeter, in inches, of the regular hexagon? |
| MBPP+ | Write a function to find frequency of each element in a flattened list of lists, returned in a dictionary. |
| Maze | Given a 2D coordinate system where both the x-axis and y-axis range from 0 to 10 (i.e., units 0, 1, .., 10). Consider a point starting at position (0,0). The goal is to move this point step by step to the target position: (8,4). During the moving, you cannot pass the following position: (1,1), (3,4), (6,2). At each step, the point may move only one unit right (r) or one unit up (u). Please provide one possible sequence of moves to reach the destination. |

blocked areas. For every question across these datasets, correctness can be assessed objectively, and multiple valid solution paths may exist. To balance cost and efficiency, we sample 100 questions from the test split of each dataset for our experiments. Table 1 shows question examples from each dataset. Further details on all three datasets are provided in Appendix A.1.

**Other Settings**   We conducted experiments on the three datasets independently. In each experiment, different LLMs were treated as independent "testers", analogous to participants in cognitive science studies, and were repeatedly prompted to solve the same questions, thereby producing the

solution set $\mathcal{S}_{q_n}$. To ensure independence in calculating problem solving performance metric, proportion of problems for which the model's first generated solution is correct (Pass@1), and solution divergence, we randomly split the 100 sampled questions into two halves: the first 50 were used to compute $\zeta_\pi$ for each model, and the remaining 50 were used to evaluate Pass@1. Since divergence requires non-empty solution sets ($|\mathcal{S}_{q_n}| > 0$), we assigned $\zeta_{q_n} = 0$ for models that failed to produce any correct solutions within the allowed trials. To control for the effect of solution set size on divergence values, we required each model to provide the same number of correct solutions per question. For models unable to generate sufficient correct solutions, we applied random oversampling from the existing solution set to match the required size. Our study encompasses a broad spectrum of LLMs, including both open-source models (e.g., Llama-3.1 (Touvron et al., 2023), Qwen-2.5 (Yang et al., 2024)) and closed-source models (e.g., GPT-4o (Bubeck et al., 2023), Claude-3.5 (Anthropic, 2024), Gemini-1.5 (Team et al., 2023)). To ensure fairness in generation, we used the same inquiry prompt across all models and relied on their default generation parameters. Additional details about the models and prompts are provided in Appendix A.2 and Appendix A.3, respectively.

### 3.3 KEY FINDINGS

We present the relationship between the Pass@1 and $\zeta_\pi$ across the three datasets in Figure 2. From the plots, we observe a consistent positive relationship between $\zeta_\pi^g$ and problem-solving performance across all three tasks, supporting our hypothesis that LLM performance is related to solution divergence. In contrast, the local metric $\zeta_\pi^l$ fails to capture this relationship on MBPP+, suggesting that $\zeta_\pi^g$ is a more reliable indicator of solution divergence. To quantify these relationships, we fit linear regression lines for each dataset and report the coefficient of determination ($R^2$), which measures the proportion of variance explained. As shown in the plots, $R^2$ values obtained with $\zeta_\pi^g$ are consistently higher than those with $\zeta_\pi^l$, further validating this observation.

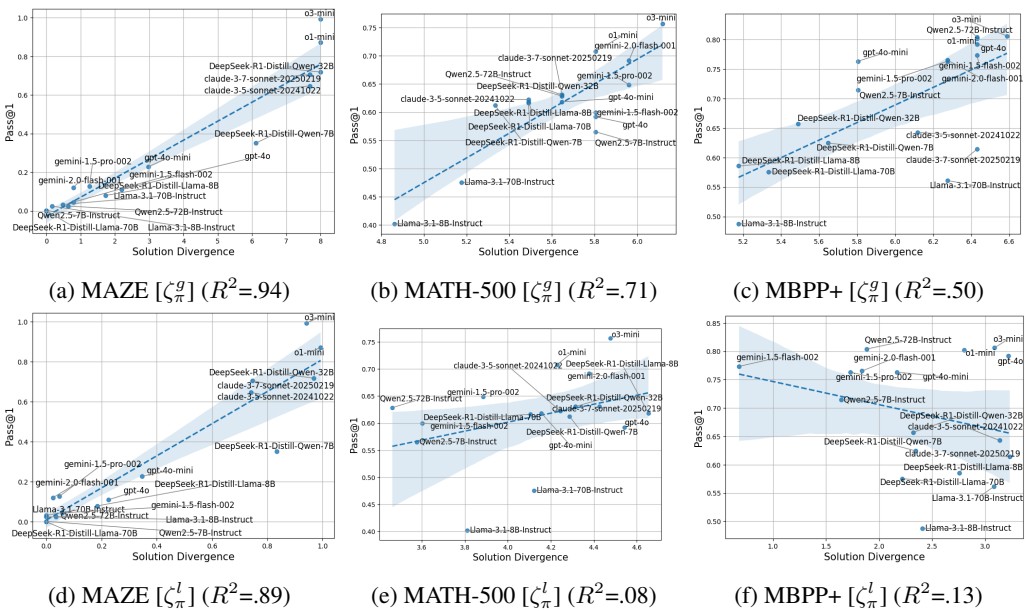

(a) MAZE [$\zeta_\pi^g$] ($R^2$=.94)     (b) MATH-500 [$\zeta_\pi^g$] ($R^2$=.71)     (c) MBPP+ [$\zeta_\pi^g$] ($R^2$=.50)

(d) MAZE [$\zeta_\pi^l$] ($R^2$=.89)     (e) MATH-500 [$\zeta_\pi^l$] ($R^2$=.08)     (f) MBPP+ [$\zeta_\pi^l$] ($R^2$=.13)

Figure 2: The Relationship of Solution Divergence ($\zeta_\pi^l$, $\zeta_\pi^g$) to Success Rate (Pass@1) in Maze, Math-500, and MBPP+ Datasets.

To further examine the alignment between human and LLM problem-solving behavior, we divide each dataset into three difficulty groups, i.e., Easy, Medium, and Hard, based on the 33rd and 66th percentiles of average success rates across all models. For each group, we compute $\zeta_\pi^g$, denoted as $\zeta_{\pi(e)}^g$, $\zeta_{\pi(m)}^g$, and $\zeta_{\pi(h)}^g$, corresponding to the easy, medium, and hard subsets, while keeping Pass@1 calculated on the mixed-difficulty questions in its original half as the performance metric. Figure 3 shows the relationship between these group-specific divergence values and Pass@1. We find that the slope $\beta$ of the fitted line is consistently steepest for $\zeta_{\pi(m)}^g$, echoing findings in cognitive science (Caviola et al., 2018) that solution divergence is most informative in the mid-difficulty range.

This alignment further suggests that divergence serves as a particularly meaningful indicator of capability under moderate problem difficulty.

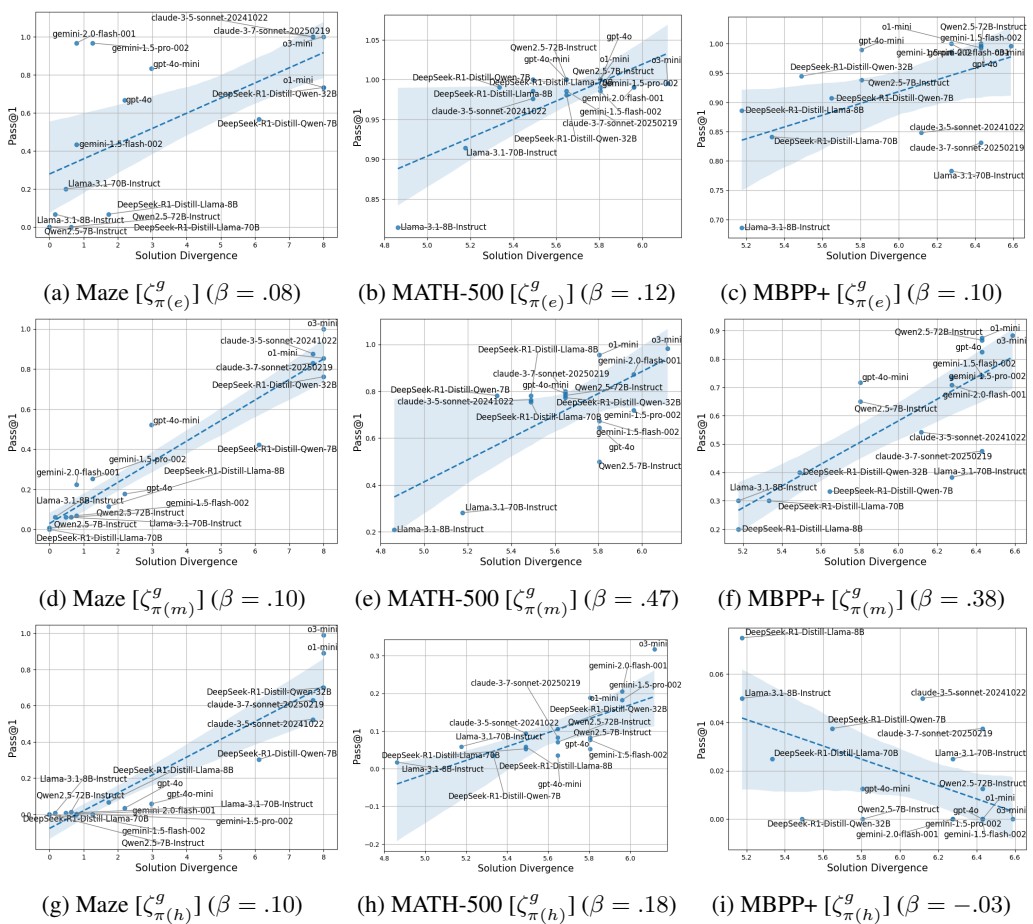

Figure 3: The Relationship of Solution Divergence ($\zeta_{\pi(e)}^{g}$, $\zeta_{\pi(m)}^{g}$, $\zeta_{\pi(h)}^{g}$) to Success Rate (Pass@1) in Maze, Math-500, and MBPP+ Datasets.

## 4 DIVERGENCE FUSED FINE-TUNING METHODS

Based on the findings in Section 3.3, we confirm a positive relationship between solution divergence ($\zeta_{\pi}^{g}$) and model problem-solving performance (Pass@1) during inference. However, the effectiveness of solution divergence as a metric remains unverified in the training stage. Inspired by recognition science studies Siegler (1998), which emphasize the benefits of fostering large repertoires of problem-solving strategies in children's education, we propose two simple yet effective approaches that integrate solution divergence into existing training paradigms, i.e., SFT and RL, to further improve LLM performance. The following sections describe each method in detail.

### 4.1 DATASET DIVERGENCE METRIC

The most straightforward way to leverage $\zeta_{\pi}$ for improving model performance during training is to use it as a criterion for data sample selection in the fine-tuning stage. The goal is to increase solution divergence by training the model on more diverse solutions. Data quality control is crucial in fine-tuning, as it directly affects both training efficiency and final performance (Shen, 2024). Building on this idea, our first proposed method for enhancing LLM problem-solving ability is to adopt $\zeta_{q_n}$ as a new metric for solution set selection. Specifically, for a given set of solutions $\mathcal{S}_{q_n}$ for the questions $q_n$, we compute its solution divergence to decide whether to add new solutions or

remove low-value ones. The decision is guided by the change in $\zeta_{q_n}^g$ before and after modification: if the metric increases, the modification is accepted; otherwise, it is rejected. In this way, incorporating solution divergence into the supervised fine-tuning process ensures higher diversity within the training dataset, which in turn enhances the model's problem-solving capability.

## 4.2 SOLUTION DIVERGENCE FUSED REWARD

Another application of the solution divergence metric is its integration into reinforcement learning (RL) training for LLMs. Recent advances in reinforcement learning algorithms, such as GRPO (Shao et al., 2024), DAPO (Yu et al., 2025), and GSPO (Zheng et al., 2025), have demonstrated the effectiveness of using group-based success rewards, evaluated over a set of generated solutions $\mathcal{S}$[1] for each question, in removing the need for a separate value model, as required in the original RLHF framework (Ouyang et al., 2022). However, these approaches focus solely on correctness-based rewards and neglect the solution divergence naturally present in group generation. To address this gap, we propose a novel divergence-augmented reward function defined as:

$$\mathcal{R}_\zeta(s_i, \mathcal{S}) = \begin{cases} \left(\frac{|\mathcal{S}_c|}{|\mathcal{S}|}\right)^\alpha \cdot \frac{\sum_{s_j \in \mathcal{S}_c} \delta(s_i, s_j)}{|\mathcal{S}_c|}, & \text{if } v(s_i) = 1, \\ -1, & \text{if } v(s_i) = 0. \end{cases} \quad (3)$$

where $s_i$ is the $i$-th generated solution, and $v(\cdot)$ is a verification function that returns 1 if $s_i$ is correct and 0 otherwise. $\mathcal{S}_c = \{s_i \mid v(s_i) = 1\} \subseteq \mathcal{S}$ is the subset of correct solutions. $\delta(\cdot)$ denotes pairwise solution divergence calculation function, $|\mathcal{S}|$ is the number of sampled solutions, and $\alpha \in \mathbb{R}$ is a scaling hyperparameter. Compared with binary correctness-based rewards, $\mathcal{R}_\zeta$ incorporates pairwise solution divergence into reward computation for correct solutions. The leading term $(|\mathcal{S}_c|/|\mathcal{S}|)$ calculates the average success rate of the solution set and serves to balance correctness and diversity: when the success ratio is low, the reward emphasizes correctness; when success is high, it shifts attention toward divergence. The hyperparameter $\alpha$ controls the sensitivity of this balance. Intuitively, while classical binary success-based rewards treat all correct answers equally, $\mathcal{R}_\zeta$ reweights correct solutions according to their divergence. In standard RL training, optimization tends to concentrate on the dominant cluster of similar solutions, giving less influence to correct but more creative or diverse answers. By reweighting samples based on solution divergence, $\mathcal{R}_\zeta$ provides a more balanced learning signal between dominant and inventive solutions, enabling the model to learn from a broader range of behaviors. By summing over all solutions, we obtain the group reward for question $q_n$:

$$\mathcal{R}_{q_n} = \sum_{s_i \in \mathcal{S}} \mathcal{R}_\zeta(s_i, \mathcal{S}) \approx \left(\frac{|\mathcal{S}_c|}{|\mathcal{S}|}\right)^{\alpha-3} \cdot \zeta_{q_n}^g + |\mathcal{S}_c| - |\mathcal{S}| \quad (4)$$

where $\zeta_{q_n}$ is the solution divergence for question $q_n$ we defined in Section 3.1. Full details of the simplification are provided in Appendix A.5. This formulation shows that the reward depends jointly on $\zeta_{q_n}$ and $|\mathcal{S}_c|$, encouraging the model not only to increase the number of correct solutions but also to diversify the solution set. For optimization, we adopt the Token-level Policy Gradient Loss proposed in DAPO, which alleviates the underweighting of long responses in the original GRPO loss. Furthermore, we remove the KL-divergence constraint to allow for broader exploration during RL training. The loss function is given by:

$$\mathcal{J}(\theta) = \mathbb{E}_{(q,a) \sim \mathcal{D}, \{s_i\}_{i=1}^{\mathcal{S}} \sim \pi_{\theta_{old}}(\cdot|q)}$$
$$\left[ \frac{1}{\sum_{i=1}^{|\mathcal{S}|} |s_i|} \sum_{i=1}^{|\mathcal{S}|} \sum_{t=1}^{|s_i|} \min\left( r_{i,t}(\theta) \hat{A}_{i,t}, \text{clip}\left( r_{i,t}(\theta), 1 - \epsilon, 1 + \epsilon \right) \hat{A}_{i,t} \right) \right] \quad (5)$$

where

---

[1] We omit the index in the following text for clarity

$$r_{i,t}(\theta) = \frac{\pi_\theta(s_{i,t}|q, s_{i,<t})}{\pi_{\theta_{old}}(s_{i,t}|q, s_{i,<t})}, \quad \hat{A}_{i,t} = \frac{\mathcal{R}_i - \text{mean}(\{\mathcal{R}_i\}_{i=1}^{|\mathcal{S}|})}{\text{std}(\{\mathcal{R}_i\}_{i=1}^{|\mathcal{S}|})} \quad (6)$$

Here, $s_{i,t}$ denotes the $t$-th token of solution $s_i$, $\theta$ is the parameter of the current policy model, and $\theta_{old}$ is the parameter of the reference model. The functions mean and std compute the mean and standard deviation of the rewards across the solution set $\mathcal{S}$. Finally, $\epsilon$ is the clipping hyperparameter that prevents excessively large updates and stabilizes RL training.

## 5 EXPERIMENT

In this section, we present the experimental details for the same three problem-solving tasks introduced in Section 3. As in the previous section, we first describe the datasets and the preparation steps applied to each. We then provide details of the models and experimental settings. Next, we report results from both SFT- and RL-based algorithms, followed by ablation studies.

### 5.1 DATASET

Details of the problems posed to LLMs are provided in Section 3.2. Here, we focus on the preparation of datasets used for subsequent SFT and RL training. For each task, we construct disjoint datasets for SFT ($\mathcal{D}_{\text{SFT}}$) and RL ($\mathcal{D}_{\text{RL}}$) by random sampling from the standard training split, ensuring $\mathcal{D}_{\text{SFT}} \cap \mathcal{D}_{\text{RL}} = \varnothing$. Dataset sizes were determined by the availability of samples, task difficulty, and computational constraints. Specifically, we use 2,000 and 1,000 questions for math, 98 and 98 for programming, and 250 and 1,000 for logical reasoning in SFT and RL training, respectively. Since our SFT experiments require diverse correct solutions per question, we employ advanced LLMs (e.g., GPT-4o, Gemini-2.5, Claude-3.5) to generate at least 10 distinct correct solutions for each. For each question, we enumerate all 4-solution subsets, compute their solution divergence $\zeta_{q_n}$, and select the subset with the highest divergence as $\mathcal{D}_{\mathcal{S}}^+$ and the one with the lowest divergence as $\mathcal{D}_{\mathcal{S}}^-$. Aggregating these across all questions yields two version training sets: a high-divergence set $\mathcal{D}_{\text{SFT}}^+$ and a low-divergence set $\mathcal{D}_{\text{SFT}}^-$. Each version SFT datasets contain 8,000, 392, and 1,000 solutions for math, programming, and logical reasoning, respectively. For validation, we sample 100, 32, and 500 examples from the validation splits of the respective datasets, which are used for both SFT and RL training. For testing, we extend the datasets from the preliminary study: the full 500-question Math-500 set for math, the same 100-question MBPP+ set for programming (as no additional test data are available), and 500 questions for Maze. Additional details on sampling procedures and prompts are provided in Appendix A.6.

### 5.2 SETTINGS

We use four representative open-source LLMs in our experiments: Llama-3.2-1B, Llama-3.1-8B, Qwen2.5-1.5B, and Qwen2.5-7B. Following the commonly adopted performance improvement pipeline, each model is first trained with the task-specific SFT dataset, and then further refined with the RL dataset for additional performance gains. For the SFT stage, we train each model on both versions of the prepared datasets, $\mathcal{D}_{\text{SFT}}^+$ and $\mathcal{D}_{\text{SFT}}^-$. In the RL stage, we further train the models using the RL dataset $\mathcal{D}_{\text{RL}}$, starting from the two SFT-trained checkpoints. In cases where the SFT results show little difference between $\mathcal{D}_{\text{SFT}}^+$ and $\mathcal{D}_{\text{SFT}}^-$, we proceed with the model trained on $\mathcal{D}_{\text{SFT}}^+$ as the initialization for RL training. For both SFT and RL training, we tune the hyper-parameters including learning rate and the solution divergence balancing factor $\alpha$. The baselines are defined as follows: for SFT, models trained on $\mathcal{D}_{\text{SFT}}^-$ serve as the baseline for comparison with $\mathcal{D}_{\text{SFT}}^+$; for RL, we replace the divergence-fused reward function $\mathcal{R}_\zeta$ with the classical binary success-based reward function $\mathcal{R}_s$. For evaluation, we report Pass@1 (success rate for the top solution) and Pass@10 (success rate within the top 10 solutions, capturing gains from solution diversity and supporting post-hoc ensembling methods such as self-consistency Wang et al. (2022)). Task instruction prompts are identical to those used in our preliminary study; other details are provided in Appendix A.7.

Table 2: Problem-solving performance (Pass@1 and Pass@10, in %) across three datasets. Results are shown for models fine-tuned with $\mathcal{D}_{\text{SFT}}^{-}$ and $\mathcal{D}_{\text{SFT}}^{+}$. The metric difference $\Delta = \mathcal{D}_{\text{SFT}}^{+} - \mathcal{D}_{\text{SFT}}^{-}$.

| Model | | Llama-3.2-1B | | | Llama-3.1-8B | | | Qwen2.5-1.5B | | | Qwen2.5-7B | | |
|---|---|---|---|---|---|---|---|---|---|---|---|---|---|
| Dataset | Metric | $\mathcal{D}_{\text{SFT}}^{-}$ | $\mathcal{D}_{\text{SFT}}^{+}$ | $\Delta$ | $\mathcal{D}_{\text{SFT}}^{-}$ | $\mathcal{D}_{\text{SFT}}^{+}$ | $\Delta$ | $\mathcal{D}_{\text{SFT}}^{-}$ | $\mathcal{D}_{\text{SFT}}^{+}$ | $\Delta$ | $\mathcal{D}_{\text{SFT}}^{-}$ | $\mathcal{D}_{\text{SFT}}^{+}$ | $\Delta$ |
| Maze | Pass@1 | 23.84 | 23.24 | -0.60 | 25.70 | 29.36 | 3.66 | 27.92 | 26.48 | -1.44 | 25.10 | 22.70 | -2.40 |
| | Pass@10 | 35.20 | 43.80 | 8.60 | 36.80 | 47.80 | 11.00 | 39.00 | 43.60 | 4.60 | 36.00 | 45.80 | 9.80 |
| Math-500 | Pass@1 | 22.88 | 25.38 | 2.50 | 38.16 | 39.24 | 1.08 | 31.78 | 32.14 | 0.36 | 43.20 | 44.78 | 1.58 |
| | Pass@10 | 40.00 | 48.60 | 8.60 | 64.20 | 72.40 | 8.20 | 53.00 | 57.40 | 4.40 | 60.80 | 69.00 | 8.20 |
| MBPP+ | Pass@1 | 26.50 | 29.60 | 3.10 | 47.00 | 47.90 | 0.90 | 40.30 | 41.30 | 1.00 | 56.00 | 54.10 | -1.90 |
| | Pass@10 | 39.00 | 51.00 | 12.00 | 70.00 | 68.00 | -2.00 | 62.00 | 63.00 | 1.00 | 65.00 | 65.00 | 0.00 |

## 5.3 MAIN RESULTS

**Dataset Divergence Metric**  Table 2 reports problem-solving performance and solution divergence across the three datasets for models fine-tuned on low- and high-divergence training samples. Models trained on $\mathcal{D}_{\text{SFT}}^{+}$ outperform those trained on $\mathcal{D}_{\text{SFT}}^{-}$ in 8 out of 12 cases for Pass@1 (mean($\Delta$) = 0.65%). The advantage is even clearer for Pass@10, where $\mathcal{D}_{\text{SFT}}^{+}$ yields higher performance in 10 out of 12 cases (mean($\Delta$) = 6.2%). These results underscore the strong influence of solution divergence in training data on SFT model performance, supporting divergence-based sample selection as an effective strategy for improving problem-solving ability. An exception is observed on the MBPP+ dataset, where performance differences between low- and high-divergence training are minimal. Examination of the training process indicates that the limited size of MBPP+ leads to rapid overfitting and early stopping, thereby reducing the benefits of divergence-based sample selection. Finally, as some values of $\Delta$ are small, we repeat the experiment over small sized model for multiple times and report their statical significance in Appendix A.8.

**Solution Divergence Fused Reward**  We now present the RL results in Table 3. When applying RL to models initialized from $\mathcal{D}_{\text{SFT}}^{-}$, the divergence-fused reward $\mathcal{R}_{\zeta}$ outperforms the binary success reward $\mathcal{R}_{s}$ on Pass@1 in 7 out of 12 cases, with an average improvement of 0.34%. More importantly, on Pass@10, models trained with $\mathcal{R}_{\zeta}$ achieve overwhelming advantages in 11 out of 12 cases, with an average improvement of 3.12% over $\mathcal{R}_{s}$. For models initialized from $\mathcal{D}_{\text{SFT}}^{+}$, we observe a similar but less pronounced trend. Specifically, $\mathcal{R}_{\zeta}$ improves Pass@1 performance in 5 out of 9 cases, though the average performance lags behind $\mathcal{R}_{s}$ by 0.38%. For Pass@10, however, $\mathcal{R}_{\zeta}$ maintains its advantage, outperforming $\mathcal{R}_{s}$ in 6 out of 9 cases with an average gain of 2.68%. These findings provide strong evidence that the divergence-fused reward is effective in enhancing the problem-solving ability of LLMs. Notably, in several cases $\mathcal{R}_{\zeta}$ yields lower Pass@1 performance but significantly better Pass@10 results. This suggests that $\mathcal{R}_{\zeta}$ encourages broader exploration of solution space, which helps models discover diverse solutions to new problems, rather than relying solely on rigid, known solutions. Together, these results confirm the effectiveness of incorporating solution divergence into the reward function. Additionally, we run multiple experiments using smaller-sized models and report their statistical significance in Appendix A.8. Finally, while our main experiments use normalized edit distance as the proxy metric for paired-solution divergence, we also evaluate a variant of $\mathcal{R}_{\zeta}$ that leverages semantic embeddings to compute divergence. We find that embedding-based divergence can further improve performance on Math-500, though it presents challenges on datasets such as MBPP+. Detailed results are reported in Appendix A.9.

## 5.4 ABLATION STUDIES

In addition to the main results, we conduct ablation studies on the Maze dataset, where task performance is highly sensitive to changes in solution divergence. This sensitivity makes Maze a useful setting for uncovering further insights into how solution divergence influences model behavior across different scenarios. In addition, because Maze is a new problem-solving task introduced in this study, using it for case analyses helps avoid the potential data-contamination issues present in the other two public benchmark datasets.

**Data Size Influence.**  We investigate the influence of SFT training data size on the relationship between solution divergence and LLM problem-solving performance. Specifically, we expand the

Table 3: Problem-solving performance (Pass@1 and Pass@10, in %) across three datasets. Results are shown for models trained with GRPO using $\mathcal{R}_s$ and $\mathcal{R}_\zeta$ as reward function, initialized from the SFT model with $\mathcal{D}_{\text{SFT}}^-$ and $\mathcal{D}_{\text{SFT}}^+$ respectively. The metric difference $\Delta = \mathcal{R}_\zeta - \mathcal{R}_s$. - denotes for skipped results due to the similar performance between $\mathcal{D}_{\text{SFT}}^-$ and $\mathcal{D}_{\text{SFT}}^+$.

| Model | | Llama-3.2-1B | | | Llama-3.1-8B | | | Qwen2.5-1.5B | | | Qwen2.5-7B | | |
|---|---|---|---|---|---|---|---|---|---|---|---|---|---|
| Dataset | Metric | $\mathcal{R}_s$ | $\mathcal{R}_\zeta$ | $\Delta$ | $\mathcal{R}_s$ | $\mathcal{R}_\zeta$ | $\Delta$ | $\mathcal{R}_s$ | $\mathcal{R}_\zeta$ | $\Delta$ | $\mathcal{R}_s$ | $\mathcal{R}_\zeta$ | $\Delta$ |
| | | | | | | | $\mathcal{D}_{\text{SFT}}^-$ | | | | | | |
| Maze | Pass@1 | 25.60 | 26.86 | 1.26 | 31.80 | 32.44 | 0.64 | 29.58 | 31.34 | 1.76 | 27.28 | 26.74 | -0.54 |
| | Pass@10 | 35.40 | 40.80 | 5.40 | 39.00 | 46.40 | 7.40 | 37.40 | 41.80 | 4.40 | 36.20 | 40.20 | 4.00 |
| Math-500 | Pass@1 | 24.52 | 25.52 | 1.00 | 41.12 | 41.50 | 0.38 | 35.48 | 36.04 | 0.56 | 46.30 | 46.26 | -0.04 |
| | Pass@10 | 41.60 | 41.20 | -0.40 | 68.00 | 68.20 | 0.20 | 54.20 | 58.80 | 4.60 | 66.40 | 68.20 | 1.80 |
| MBPP+ | Pass@1 | 33.40 | 33.40 | 0.00 | 50.70 | 49.10 | -1.60 | 48.10 | 47.70 | -0.40 | 59.70 | 60.70 | 1.00 |
| | Pass@10 | 44.00 | 50.00 | 6.00 | 63.00 | 63.00 | 0.00 | 59.00 | 63.00 | 4.00 | 67.00 | 71.00 | 4.00 |
| | | | | | | | $\mathcal{D}_{\text{SFT}}^+$ | | | | | | |
| Maze | Pass@1 | 32.60 | 30.94 | -1.66 | 37.82 | 34.26 | -3.56 | 31.06 | 30.66 | -0.40 | 26.64 | 27.34 | 0.70 |
| | Pass@10 | 43.20 | 47.60 | 4.40 | 43.00 | 54.40 | 11.40 | 43.40 | 46.40 | 3.00 | 33.00 | 43.80 | 10.80 |
| Math-500 | Pass@1 | 26.92 | 27.44 | 0.52 | 43.58 | 39.68 | -3.90 | 34.42 | 34.78 | 0.36 | 47.50 | 50.22 | 2.72 |
| | Pass@10 | 50.20 | 49.20 | -1.00 | 71.80 | 69.80 | -2.00 | 57.60 | 62.40 | 4.80 | 69.00 | 70.80 | 1.80 |
| MBPP+ | Pass@1 | 37.90 | 38.60 | 0.70 | - | - | - | - | - | - | - | - | - |
| | Pass@10 | 58.00 | 57.00 | -1.00 | - | - | - | - | - | - | - | - | - |

number of unique questions in $\mathcal{D}_{\text{SFT}}^+$ and $\mathcal{D}_{\text{SFT}}^-$ from 250 to 1,000, thereby increasing the dataset sizes from 1,000 to 4,000. This yields two new datasets: $\mathcal{D}_{\text{SFT}}^{++}$ and $\mathcal{D}_{\text{SFT}}^{--}$. Figure 4 reports model performance across all four SFT datasets. The results show that training with more samples improves overall performance, while also widening the gap between low- and high-divergence datasets, further highlighting the benefit of incorporating solution divergence in SFT training.

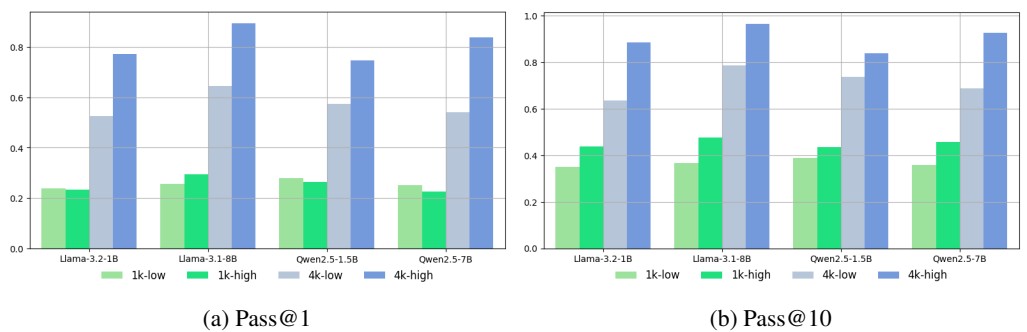

(a) Pass@1                    (b) Pass@10

Figure 4: Problem-solving performance (Pass@1 and Pass@10, in %) of models trained on $\mathcal{D}_{\text{SFT}}^-$ (1k-low), $\mathcal{D}_{\text{SFT}}^{--}$ (4k-low), $\mathcal{D}_{\text{SFT}}^+$ (1k-high), and $\mathcal{D}_{\text{SFT}}^{++}$ (4k-high) for solving problems in Maze.

**Performance and Divergence Balanced Reward.** In Section 4.2, we introduced the hyperparameter $\alpha$ to balance solution divergence and problem-solving performance during RL training. We experiment with $\alpha \in \{2, 3, 4\}$, which scales the divergence term to be inversely proportional, constant, or directly proportional to the success rate. As shown in Figure 5, $\alpha = 4$ yields the best performance in 5 out of 8 Pass@1 cases and 6 out of 8 Pass@10 cases. This suggests that it is generally beneficial to limit the influence of divergence when problem-solving performance is low, and to amplify it as performance improves. This finding aligns with cognitive science theory (Siegler, 1998), which emphasizes first developing a correct strategy and only then expanding to diverse alternatives.

**Generation Temperature Tuning.** Tuning the generation temperature is a common technique for adjusting a model's output style without retraining. In general, higher temperatures encourage more diverse and creative generations, which can lead to improvements in Pass@10 on problem-solving benchmarks. To compare the effect of temperature tuning with our proposed solution-

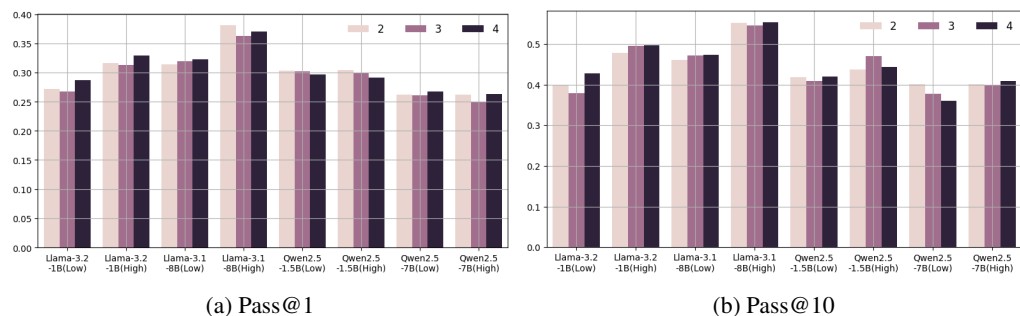

(a) Pass@1          (b) Pass@10

Figure 5: Problem-solving performance (Pass@1 and Pass@10, in %) of models trained on $\alpha = 2, 3, 4$ for solving problems in Maze.

Table 4: Problem-solving performance (Pass@1 and Pass@10, in %) on Maze using different generating temperatures. Results are shown for models fine-tuned with $\mathcal{D}_{\text{SFT}}^-$, $\mathcal{D}_{\text{SFT}}^+$, $\mathcal{R}_s$ and $\mathcal{R}_\zeta$.

| Model | | Llama-3.2-1B | | | | | | Qwen2.5-1.5B | | | | | |
|---|---|---|---|---|---|---|---|---|---|---|---|---|---|
| Method | | $\mathcal{D}_{\text{SFT}}^-$ | | $\mathcal{D}_{\text{SFT}}^+$ | $\mathcal{R}_s$ | | $\mathcal{R}_\zeta$ | $\mathcal{D}_{\text{SFT}}^-$ | | $\mathcal{D}_{\text{SFT}}^+$ | $\mathcal{R}_s$ | | $\mathcal{R}_\zeta$ |
| Temperature | | 0.9 | 1.2 | 0.6 | 0.9 | 1.2 | 0.6 | 0.9 | 1.2 | 0.7 | 0.9 | 1.2 | 0.7 |
| Maze | Pass@1 | 22.54 | 21.08 | 23.40 | 25.10 | 23.58 | 26.86 | 27.45 | 27.04 | 26.48 | 29.09 | 28.85 | 31.34 |
| | Pass@10 | 40.00 | 41.66 | 43.80 | 38.20 | 41.68 | 40.80 | 39.88 | 42.84 | 43.60 | 38.32 | 41.12 | 41.80 |
| | Average | 31.27 | 31.37 | 33.60 | 31.65 | 32.63 | 33.83 | 33.67 | 34.94 | 35.04 | 33.71 | 34.99 | 36.57 |
| Math-500 | Pass@1 | 22.66 | 22.20 | 25.38 | 23.31 | 21.34 | 25.52 | 31.75 | 31.10 | 32.14 | 35.69 | 35.28 | 36.04 |
| | Pass@10 | 41.32 | 42.12 | 48.60 | 43.30 | 43.60 | 42.20 | 54.56 | 55.84 | 57.40 | 56.60 | 59.00 | 58.80 |
| | Average | 31.99 | 32.16 | 36.99 | 33.31 | 32.47 | 33.86 | 43.16 | 43.47 | 44.77 | 46.15 | 47.14 | 47.42 |
| MBPP+ | Pass@1 | 24.02 | 21.42 | 29.60 | 33.94 | 32.00 | 33.40 | 40.12 | 34.80 | 41.30 | 44.23 | 27.02 | 47.70 |
| | Pass@10 | 41.00 | 53.00 | 51.00 | 42.92 | 49.37 | 50.00 | 62.40 | 65.80 | 63.00 | 64.38 | 64.38 | 63.00 |
| | Average | 32.51 | 37.21 | 40.30 | 38.43 | 40.69 | 41.70 | 51.26 | 50.30 | 52.15 | 54.31 | 45.70 | 55.35 |

divergence–based training, we conduct the following experiments. For the SFT stage, we take the model trained on $\mathcal{D}_{\text{SFT}}^-$ and generate solutions using two additional temperatures: 0.9, and 1.2. We compare these results against the model trained on $\mathcal{D}_{\text{SFT}}^+$, evaluated with the default temperature. For the RL stage, we similarly evaluate the model trained with $\mathcal{R}_s$ at different temperatures and compare it to the model trained with $\mathcal{R}_\zeta$ under the default temperature. Results are reported in Table 4. Because both Pass@1 and Pass@10 are important, we additionally report their average. From the table, we observe that increasing the temperature improves the Pass@10 of models trained on $\mathcal{D}_{\text{SFT}}^-$ and $\mathcal{R}_s$, but typically reduces Pass@1. When comparing the averages of the two metrics, models trained with $\mathcal{D}_{\text{SFT}}^+$ and $\mathcal{R}_\zeta$ consistently maintain the best performance. These results suggest that solution-divergence–based training provides more robust overall gains than temperature tuning alone. In addition to the Maze, we also experiment result experiments with both the MBPP+ and Math-500 datasets and the consistent observation can be observed.

# 6 CONCLUSION

In this paper, we investigate an underexplored direction for enhancing the problem-solving performance of LLMs: solution divergence, defined as the presence of multiple viable solutions to a single problem. Our preliminary study empirically demonstrates a positive relationship between solution divergence and model performance. Building on this insight, we introduce two methods, the dataset divergence metric and a divergence-fused reward, to augment existing SFT and RL algorithms. Comprehensive experiments across three representative problem-solving tasks in the logical reasoning, mathematics, and programming domains confirm the effectiveness of leveraging solution divergence to improve LLM performance. These findings highlight the potential of solution divergence as a valuable training signal and open new avenues for future research on harnessing diversity in solutions to strengthen LLM problem-solving capabilities.

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

# A APPENDIX

## A.1 PRELIMINARY STUDY DATASET DETAILS

We conduct our preliminary study on the relationship between LLMs' solution divergence ($\zeta_\pi$) and problem-solving performance (Pass@1) across three datasets: Math-500, MBPP+, and Maze.

**Math-500.** This dataset is a high-quality subset of Math, a widely used benchmark for evaluating LLMs' mathematical problem-solving ability. Unlike the full Math dataset, which includes standard train/validation/test splits, Math-500 contains 500 correctness-verified questions sampled from the original test split. The dataset spans seven categories and provides ground-truth answers, with most questions also annotated with difficulty levels (1–5). For efficiency and to control API costs when querying closed-source models, we randomly sampled 100 questions from Math-500 for our experiments.

**MBPP+.** MBPP+ is a refined version of the MBPP benchmark, designed to evaluate LLMs' ability to solve Python programming tasks. Each problem specifies a function signature and requires a correct implementation that passes the associated test cases. Compared to MBPP, MBPP+ improves test robustness by expanding coverage of edge cases; a small number of ambiguous problems from MBPP were removed. In total, MBPP+ contains 378 problems, aligned with the original MBPP train/validation/test split. For consistency with Math-500, we randomly sampled 100 test problems to form our preliminary programming dataset. Both Math-500 and MBPP+ are accessed via the Hugging Face datasets library[2].

**Maze.** Maze is a new logical reasoning dataset introduced in this paper. Each problem asks an LLM to find a viable path from a fixed start point (0,0) to a random goal point within a $10 \times 10 grid$. To increase difficulty, we add a blocking set $\mathcal{B}$, where $|\mathcal{B}| < 100$, with blocked coordinates sampled uniformly at random. Each problem instance has a distinct blocking configuration. Following the setup for Math-500 and MBPP+, we generated 100 Maze problems for the preliminary study (see Table 1 for an illustration).

**Verification.** To verify solution correctness across tasks, we combine existing open-source verification tools with additional rules tailored to each dataset. For Math-500, we use the open-source math_verify package[3]. This tool processes full solution strings, extracts the marked answer, and handles common numerical-equivalence cases with high accuracy. Since our prompts instruct the model to generate step-by-step solutions, we extract the final answer from the last step and pass it, along with the ground-truth answer, to the verifier. For MBPP+, we execute each generated function against the dataset's official Python test cases. We cap the runtime of each test at 200 ms; if execution raises an error or exceeds the time limit, the solution is marked incorrect. For Maze, we reconstruct the complete movement trajectory from the generated solution. A solution is judged correct only if the final position reaches the goal and all intermediate positions avoid any blocked coordinates in $\mathcal{B}$.

## A.2 PRELIMINARY STUDY MODELS

To ensure the robustness of our conclusion, we comprehensively select 17 representative LLMs from both open- and close-sourced ones. Below, we list them by their series names as follows: o1-mini, o1-mini, gpt-4o, gpt-4o-mini, claude-3-7-sonnet-20250219, claude-3-5-sonnet-20241022, gemini-1.5-pro-002, gemini-1.5-flash-002, gemini-2.0-flash-001, Llama-3.1-70B-Instruct, Llama-3.1-8B-Instruct, Qwen2.5-72B-Instruct, Qwen2.5-7B-Instruct, DeepSeek-R1-Distill-Qwen-32B, DeepSeek-R1-Distill-Qwen-7B, DeepSeek-R1-Distill-Llama-70B, DeepSeek-R1-Distill-Llama-8B. For all the close-sourced ones, we implement based on the huggingface Transformer package[4] and we use the default generation configuration for each model inference. For the open-sourced LLMs. we send request to official API endpoints for the results.

---

[2]https://huggingface.co/docs/datasets/index

[3]https://github.com/huggingface/Math-Verify

[4]https://huggingface.co/docs/transformers/en/index

## A.3 PRELIMINARY STUDY TASK PROMPTS

Below, we present the query prompts for each task. To improve the accuracy of approximating pairwise divergence $\delta_{i,j}$ via string edit distance, we explicitly include output-format instructions in each prompt, requiring LLMs to produce solutions in a standardized style. This design reduces formatting noise and expression redundancy, ensuring cleaner comparisons during our experiments.

> Given a 2D coordinate system where both the x-axis and y-axis range from 0 to 10 (i.e., units 0, 1, .., 10). Consider a point starting at position (0,0). The goal is to move this point step by step to the target position: {target}. During the moving, you cannot pass the following position: {forbid}. At each step, the point may move only one unit right (r) or one unit up (u). Please provide one possible sequence of moves to reach the destination. The output format should follow this pattern: $\boxed{s \to r \to u \to r \to \cdots \to e}$, where s indicates the start of the path, and e indicate the end of the path. The steps in between consist only of r and u characters. If there is no viable path to make the point move to the target position, output as: $\boxed{\times}$; Do not solve this problem using code or external tools, and avoid including any form of validation or result verification at the end of your response.

Figure 6: The example prompt we used to solve the problems of Maze dataset. {target} and {forbid} are the placeholder for the destination point and block points set, respectively.

> Please provide a step-by-step solution and final answer to the following question. Avoid redundant steps, such as restating information from the question or listing pure calculations as independent steps. Do not include validation or verification at the end of the solution.
> Question: {question}
> Format your response as follows:
> Step-by-Step Solution: Step 1. ... Step 2. ... ... Step N. ...
> Final Answer: $\boxed{\text{XXX}}$
> (Replace XXX with the final computed value.)

Figure 7: The example prompt we used to solve the problems of Math-500 dataset. {question} is the placeholder for the question stem text.

> Complete the Python function below to fulfill the request below. Please avoid using try, except, or raise statements in your implementation, and focus on achieving the intended functionality. Request: {request} Return your complete function in the following format:
> ```python
> {function}
>    # start to complete the function here
> ```
>
> (Replace the comment with your actual implementation.)

Figure 8: The example prompt we used to solve the problems of MBPP+. {request} is the placeholder for the problem descriptions and {function} is the function name required by the execution of the following test cases.

## A.4 ADDITIONAL PRELIMINARY STUDY RESULTS

In this section, we present scatter plots of the local-focused solution divergence metric ($\zeta_\pi^l$) against LLM problem-solving performance (Pass@1) across the three difficulty subsets (Figure 9). From the figure, we observe that the divergence for the medium-difficulty subset ($\zeta_{\pi,m}^l$) exhibits the steepest slope compared to the easy and hard subsets in both the Maze and Math-500 datasets. However, the

MBPP+ dataset shows some inconsistencies, highlighting that the local-focused divergence metric is less reliable than the global-focused metric in capturing the relationship between solution divergence and problem-solving performance.

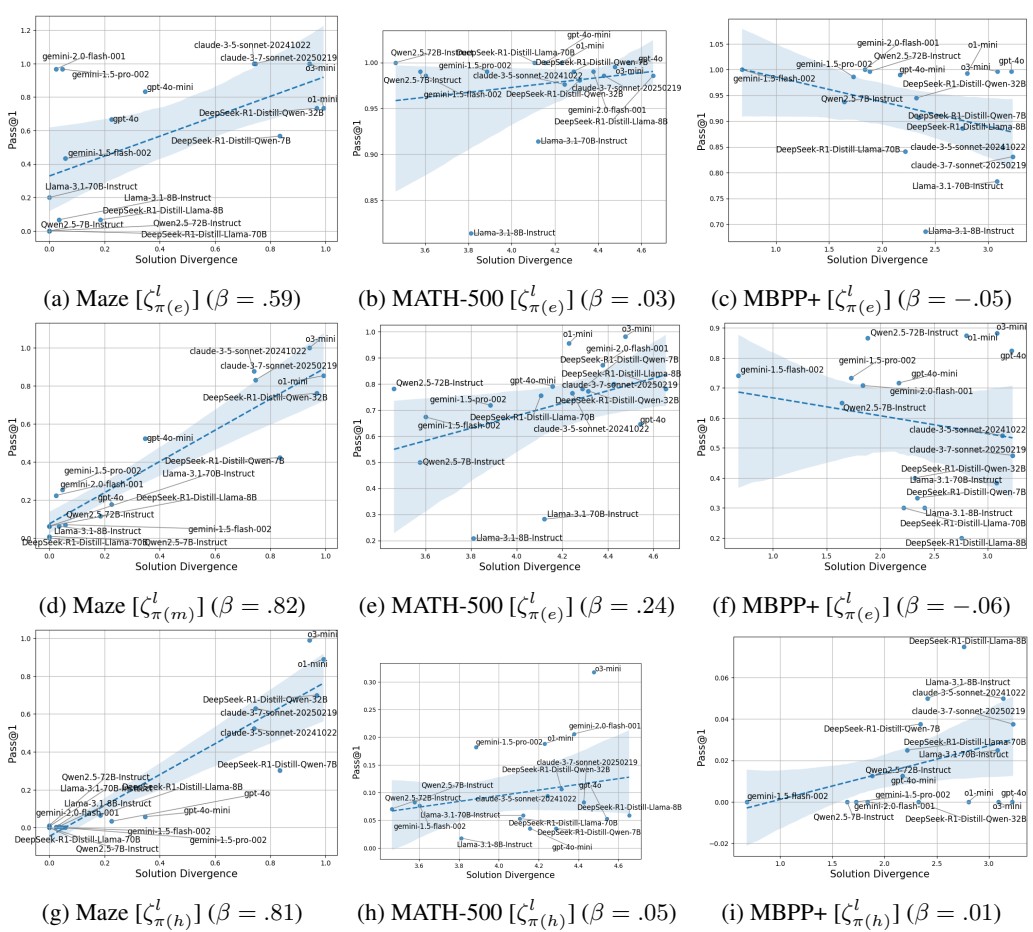

(a) Maze $[\zeta^l_{\pi(e)}]$ ($\beta = .59$)  (b) MATH-500 $[\zeta^l_{\pi(e)}]$ ($\beta = .03$)  (c) MBPP+ $[\zeta^l_{\pi(e)}]$ ($\beta = -.05$)

(d) Maze $[\zeta^l_{\pi(m)}]$ ($\beta = .82$)  (e) MATH-500 $[\zeta^l_{\pi(e)}]$ ($\beta = .24$)  (f) MBPP+ $[\zeta^l_{\pi(e)}]$ ($\beta = -.06$)

(g) Maze $[\zeta^l_{\pi(h)}]$ ($\beta = .81$)  (h) MATH-500 $[\zeta^l_{\pi(h)}]$ ($\beta = .05$)  (i) MBPP+ $[\zeta^l_{\pi(h)}]$ ($\beta = .01$)

Figure 9: The Relationship of Solution Divergence to Success Rate (Pass@1) in MAZE, MATH-500, and MBPP-Plus Problem-Solving Datasets.

## A.5 DIVERGENCE FUSED REWARD SIMPLIFICATION

In Section 4.2, we presented the simplified form of the reward function in Eq. 4. For completeness, we provide the full derivation here. We start from the original formulation:

$$\mathcal{R}_n = \sum_{s_i \in \mathcal{S}} \mathcal{R}_\zeta(s_i, \mathcal{S})$$

Substituting Eq. 3 into Eq. 4, we have

$$\mathcal{R}_n = \sum_{s_i \in \mathcal{S}_c} \left(\frac{|\mathcal{S}_c|}{|\mathcal{S}|}\right)^\alpha \frac{\sum_{s_j \in \mathcal{S}_c} \delta(s_i, s_j)}{|\mathcal{S}_c|} - \sum_{s_i \notin \mathcal{S}_c} 1$$

$$= \left(\frac{|\mathcal{S}_c|}{|\mathcal{S}|}\right)^{\alpha-1} \cdot \frac{1}{|\mathcal{S}|} \sum_{s_i \in \mathcal{S}_c} \sum_{s_j \in \mathcal{S}_c} \delta(s_i, s_j) - \left(|\mathcal{S}| - |\mathcal{S}_c|\right). \tag{7}$$

Based on the Eq. 2, we have:

$$\zeta_{q_n}^g = M - \frac{1}{M} \sum_{i=1}^{M} \lambda_i = M - \frac{1}{M}\text{tr}(\Lambda) = M - \frac{1}{M}\text{tr}(L) = M - \frac{1}{M}\text{tr}(D - A)$$

$$= M - \frac{1}{M}\text{tr}(D) = M - \frac{1}{M} \sum_{s_i \in \mathcal{S}} \sum_{s_j \in \mathcal{S}} (1 - \delta(s_i, s_j))$$

By definition $|\mathcal{S}| = M$, thus we have:

$$\zeta_{q_n}^g = |\mathcal{S}| - \frac{1}{|\mathcal{S}|} \sum_{s_i \in \mathcal{S}} \sum_{s_j \in \mathcal{S}} (1 - \delta(s_i, s_j)) = \frac{1}{|\mathcal{S}|} \sum_{s_i \in \mathcal{S}} \sum_{s_j \in \mathcal{S}} \delta(s_i, s_j) \tag{8}$$

To be noticed, if all the solutions in $\mathcal{S}$ are correct, then we will have $|\mathcal{S}| = |\mathcal{S}_|$ and the leading term in Eq. 4 will always be the constant 1, and the expression won't be influenced by $\alpha$, thus here we only consider about when $|\mathcal{S}_c| < |\mathcal{S}|$. In Section 3.2, we mention we will conduct the random over-sampling over $\mathcal{S}_c$ to match the size of relation graph $|\mathcal{G}| = |\mathcal{S}|$. Suppose we sample all the solutions in $\mathcal{S}_c$ for $k$ times samples to fulfill the requests, and $k = |\mathcal{S}|/|\mathcal{S}_c|$. We can have:

$$\frac{\sum_{s_i \in \mathcal{S}} \sum_{s_j \in \mathcal{S}} \delta(s_i, s_j)}{\sum_{s_i \in \mathcal{S}_c} \sum_{s_j \in \mathcal{S}_c} \delta(s_i, s_j)} \approx \frac{P(|\mathcal{S}|, 2) - |\mathcal{S}_c| \cdot P(k, 2)}{P(|\mathcal{S}_c|, 2)} = \frac{|\mathcal{S}|(|\mathcal{S}| - 1) - |\mathcal{S}_c| \cdot k(k - 1)}{|\mathcal{S}_c|(|\mathcal{S}_|| - 1)}$$

$$= \frac{|\mathcal{S}|(|\mathcal{S}| - 1) - |\mathcal{S}| \cdot (\frac{|\mathcal{S}|}{|\mathcal{S}_c|} - 1)}{|\mathcal{S}_c|(|\mathcal{S}_c| - 1)} = \frac{|\mathcal{S}|^2}{|\mathcal{S}_c|^2}$$

where $P(\cdot, \cdot)$ denotes for permutation operation. Plug it back to Eq. 8, $\zeta_{q_n}^g$ can be expressed as:

$$\zeta_{q_n}^g \approx \frac{1}{|\mathcal{S}|} \cdot \frac{|\mathcal{S}|^2}{|\mathcal{S}_c|^2} \sum_{s_i \in \mathcal{S}_c} \sum_{s_j \in \mathcal{S}_c} \delta(s_i, s_j)$$

Then, if we do a simply conversion to the Eq. 7, we can have:

$$\mathcal{R}_n = \left(\frac{|\mathcal{S}_c|}{|\mathcal{S}|}\right)^{\alpha-3} \cdot \frac{1}{|\mathcal{S}|} \cdot \frac{|\mathcal{S}|^2}{|\mathcal{S}_c|^2} \sum_{s_i \in \mathcal{S}_c} \sum_{s_j \in \mathcal{S}_c} \delta(s_i, s_j) - (|\mathcal{S}| - |\mathcal{S}_c|)$$

$$\approx \left(\frac{|\mathcal{S}_c|}{|\mathcal{S}|}\right)^{\alpha-3} \cdot \zeta_{q_n}^g + |\mathcal{S}_c| - |\mathcal{S}|$$

which completes the proof.

## A.6 EXPERIMENT DATA PREPARATION

In this section, we describe the preparation of training samples for the SFT and RL experiments. For the math problem-solving task, we sample questions from the original Math dataset, since Math-500 only provides a test split. Each sampled question is then fed into the solving prompt (Figure 10), where the LLM is instructed to generate four diverse solutions. By aggregating responses from multiple LLMs, we construct a candidate solution set for divergence-based selection. We enumerate all possible 4-solution combinations, compute the question-level solution divergence, and retain the sets with the highest and lowest divergence values to form the two SFT datasets for Math.

Following the same procedure, we prepare the SFT dataset for MBPP+ using the diverse-solution generation prompt in Figure 11. For Maze, since all viable paths can be enumerated via brute-force search, we directly generate solutions programmatically and randomly sample 10 solutions per question for downstream processing.

Provide four distinct solutions to the single given question. A reference solution is provided for guidance, but your solutions must be different from the reference. Each solution must be step-by-step and use a different method from the others. Avoid redundant steps (e.g., restating the problem or listing bare arithmetic as separate steps). Do not include validation or verification at the end.
Question {question}
Solution {solution}
Format your response exactly as follows:
Solution 1
Step 1. …  Step 2. …  Step 3. …
Final Answer: $\boxed{\times}$
Solution 2
Step 1. …  …
Final Answer: $\boxed{\times}$
Solution 3
Step 1. …  …
Final Answer: $\boxed{\times}$
Solution 4
Step 1. …  …
Final Answer: $\boxed{\times}$
Now, please start to respond.

Figure 10: Example prompt used to generate diverse solutions for SFT training questions of Math dataset.

Complete the four distinct Python functions below to fulfill the request described in the comments. A reference implementation is provided for guidance, but your solutions must be different from the reference. Each function should employ a unique approach, and none should rely on try, except, or raise statements. Focus on achieving the intended functionality through alternative methods.

```python
# {request}
# Reference implementation
{solution}
```

Return your complete function in the following format:

```python
# function 1
{function}

# function 2
{function}

# function 3
{function}

# function 4
{function}
```

Now, please start to respond.

Figure 11: Example prompt used to generate diverse solutions for SFT training questions of MBPP+ dataset.

Table 5: Problem-solving performance (Pass@1 and Pass@10, in %) on Maze, Math-500, and MBPP+ with multiple runs. Results are shown for models fine-tuned with $\mathcal{D}_{\text{SFT}}^-$, $\mathcal{D}_{\text{SFT}}^+$, $\mathcal{R}_s$, and $\mathcal{R}_\zeta$. For each metric, we report the mean and std, and statistically significant gaps are marked with *.

| Model | | Llama-3.2-1B | | | | | | Qwen-2.5-1.5B | | | | | |
|---|---|---|---|---|---|---|---|---|---|---|---|---|---|
| Dataset | Metric | $\mathcal{D}_{\text{SFT}}^-$ | $\mathcal{D}_{\text{SFT}}^+$ | $\Delta$ | $\mathcal{R}_s$ | $\mathcal{R}_\zeta$ | $\Delta$ | $\mathcal{D}_{\text{SFT}}^-$ | $\mathcal{D}_{\text{SFT}}^+$ | $\Delta$ | $\mathcal{R}_s$ | $\mathcal{R}_\zeta$ | $\Delta$ |
| Maze | Pass@1 | 23.55±0.41 | 23.31±0.32 | -0.24 | 25.32±0.24 | 26.57±0.37 | 1.25* | 27.81±0.15 | 26.77±0.47 | -1.04* | 29.73±0.25 | 30.11±0.43 | 0.38 |
| | Pass@10 | 34.71±0.74 | 44.71±1.09 | 10.0* | 34.88±0.83 | 40.91±0.68 | 6.03* | 39.19±0.76 | 43.45±0.84 | 4.26* | 37.15±0.55 | 42.35±0.73 | 5.20* |
| Math-500 | Pass@1 | 22.78±0.16 | 25.55±0.54 | 2.77* | 24.42±0.13 | 25.65±0.21 | 1.23* | 31.65±0.39 | 31.98±0.23 | 0.33 | 35.62±0.15 | 36.55±0.33 | 0.82* |
| | Pass@10 | 40.25±0.87 | 47.65±1.05 | 7.40* | 41.84±0.90 | 41.45±1.28 | -0.39 | 51.50±1.36 | 57.28±1.43 | 5.78* | 53.90±1.66 | 57.65±0.99 | 3.75* |
| MBPP+ | Pass@1 | 26.35±0.41 | 28.85±1.50 | 2.50* | 33.35±0.82 | 33.45±0.47 | 0.10 | 40.50±0.63 | 40.75±1.58 | 0.25 | 47.55±0.71 | 48.25±0.53 | 0.70* |
| | Pass@10 | 39.15±1.30 | 50.75±1.79 | 11.6* | 44.50±0.48 | 49.52±2.28 | 5.02* | 63.25±2.19 | 63.55±1.52 | 0.30 | 58.50±1.23 | 62.84±0.90 | 4.34* |

Table 6: Problem-solving performance (Pass@1 and Pass@10, in %) on Math-500 and MBPP+ datasets, using different pair-wised solution divergence calculation ways. Results are shown for models fine-tuned with the divergence introduced reward function $\mathcal{R}_\zeta$.

| Metric | Math-500 | | | | | | MBPP+ | | | | | |
|---|---|---|---|---|---|---|---|---|---|---|---|---|
| | Llama-3.2-1B | | | Qwen2.5-1.5B | | | Llama-3.2-1B | | | Qwen2.5-1.5B | | |
| | Edit | Embedding | $\Delta$ | Edit | Embedding | $\Delta$ | Edit | Embedding | $\Delta$ | Edit | Embedding | $\Delta$ |
| Pass@1 | 25.52 | 25.63 | 0.11 | 36.04 | 37.45 | 1.41 | 33.40 | 33.53 | 0.13 | 47.7 | 49.46 | 1.76 |
| Pass@10 | 41.20 | 42.28 | 1.08 | 58.80 | 60.47 | 1.67 | 50.00 | 47.2 | -2.80 | 63.00 | 58.70 | -4.30 |

## A.7 EXPERIMENT SETTING DETAILS

In this section, we introduce the detailed training settings used for our SFT and RL experiment. For both SFT and RL training, we tune the learning rate from $\{2\times10^{-5}, 1\times10^{-5}, 8\times10^{-6}, 5\times10^{-6}\}$, divergence balancing parameter $\alpha \in \{0, 1, 2, 3, 4, 5\}$ and set the global batch size to 64. During RL training, we fix the solution set size to $|\mathcal{S}| = 8$ per question and set the clipping parameter $\epsilon = 0.2$. Both SFT and RL stages are trained for 10 epochs using the Adam optimizer with $\beta_1 = 0.9$ and $\beta_2 = 0.999$. During the training, the best performed model on the validation dataset is saved. We implement all the training with the Huggingface TRL packages [5].

## A.8 EXPERIMENT WITH REPEATED RESULTS

As shown in Table 2 and Table 3, the performance gaps between models on metrics such as Pass@1 and Pass@10 narrow in absolute terms. To assess whether these differences remain statistically meaningful, we run inference on the same set of questions 10 times for each trained SFT and RL model using the default decoding settings, and apply a Student's t-test to evaluate the significance of the average performance differences. The resulting means and standard deviations are reported in Table 5, with statistically significant gaps ($p < 0.05$) marked by an asterisk. From the table, we observe that in most cases the performance differences between models with varying solution divergences remain significant, only 7 out of 24 comparisons are not, demonstrating the consistent influence of solution divergence.

## A.9 EXPERIMENT WITH SEMANTIC EMBEDDINGS

In this section, we conduct the experiment to examine how alternative divergence metrics might influence our findings, Specifically, we employed the "text-embedding-3-small" model from the OpenAI API to encode the solutions into embedding vectors and computed pairwise solution divergence using cosine similarity. We incorporated this embedding-based divergence into the RL training phase and evaluated its effect on two models: Llama-3.2-1B and Qwen2.5-1.5B. As shown in Table 6, the embedding consistently improves performance on the math-500 dataset, whereas the improvements on MBPP+ are less pronounced. We hypothesize that this difference arises from the mismatch between natural language and programming language: step-by-step math solutions resemble human-written text, making them more compatible with the capabilities of the embedding model, whereas program code is less aligned with its embedding space.

---

[5] https://huggingface.co/docs/trl/en/index

