# OpenReview forum: "Exploring Solution Divergence and Its Effect on Large Language Model Problem Solving"
_ICLR.cc/2026/Conference — ICLR 2026 Conference Withdrawn Submission_

### Official Review · Reviewer_i9wi · 2025-10-31

**Soundness:** 3
**Presentation:** 4
**Contribution:** 2
**Rating:** 4
**Confidence:** 3

**Summary:**

The paper argues that the diversity of solutions generated by a language model to a given problem is an important indicator of the problem solving ability of the model. It proposes a way to measure, and a way to improve, the solution diversity in models (for both supervised and reinforcement based fine-tuning) based on (the eigenvalues of) the Laplacian of a graph constructed from the solutions generated by a model. It also provides experimental evidence for the benefits of doing so.

**Strengths:**

The paper is exceptionally well written and very easy to follow. It investigates solution diversity backed with a strong motivation from cognitive science. It proposes an elegant mechanism to measure solution diversity using the Laplacian of a graph constructed from multiple solutions drawn from a model. And it shows that their measure of solution diversity can be beneficial in training models.

**Weaknesses:**

The approach is explored and evaluated on three simple reasoning tasks (math-500, mbpp, and a newly introduced maze task), represented by a sample of 100 test cases each. I am not sure how stable the findings are if one would broaden that set (or even re-ran the experiment on a different set of 100 test cases). This concern is corroborated by the fact that there is no significant performance improvement from leveraging solution divergence in one of the three kinds of task (mbpp) in Section 5.3.

Solution divergence has the most significant positive impact in the Pass@10 evaluation setting (vs. Pass@1 where there is much less of an impact). I am wondering whether this setting obviously benefits from more diverse solutions, and how significance the overall finding therefore is.

A few additional concerns are phrased as questions below.

**Questions:**

(minor comment) The text in Figure 2 is extremely hard to read. I don’t know how one would fix that, but perhaps not labelling every single model and showing just a few ones with larger font-size would work.

What is N in the experiment shown in Figure 2?

One could imagine many ways to incorporate solution divergence into RL rewards, and the formulation in Section 4.2 seems like one somewhat arbitrary way to do so. It would be nice to understand how this particular formulation (over any potential other ways) is motivated.

(minor comment) Wouldn’t “solution diversity” a more fitting term that “solution divergence”?

(line 360) “we tune the hyper-parameters including learning rate…”. How were they tuned and based on what criterion and what data?

(line 365) It is mentioned that self-consistency (Wang et al. 2022) can be leveraged. Was it used, and if so, how?

In Tables 2 and 3, data with high solution divergence is compared to data with low solution divergence. Shouldn’t there be a comparison to data with “typical” solution divergence, which would be simply a sample drawn from the model.

Further, as a control study, how would varying solution divergence of the training data by simply varying temperature, rather than based on the proposed method, compare?

There seems to be a typo in the sentence beginning in line 430.

---

> ### Author Response · Authors · 2025-11-27
>
> We appreciate your detailed comments on our paper and to resolve your current concerns to the paper, we we provide the additional experiment results and explains to some unclear part in the paper. Hope it can be helpful to you.
>
> __Q1. I am not sure how stable the findings are if one would broaden that set (or even re-ran the experiment on a different set of 100 test cases). This concern is corroborated by the fact that there is no significant performance improvement from leveraging solution divergence in one of the three kinds of task (mbpp) in Section 5.3.__
>
> A1. Thank you for raising the concern regarding performance robustness on the MBPP+ dataset. To further assess and demonstrate the stability of our conclusions, we re-split the dataset using a different random seed and re-ran all SFT and RL experiments on two models (Llama-3.2-1.5B and Qwen-2.5-1.5B) using the newly generated split. Our results show that, although the absolute performance values vary across splits, the performance gap between small- and large-divergence settings remains consistent, reinforcing the original conclusion presented in the paper.
>
> |  Metric | Llama-3.2-1B |       |       | Qwen2.5-1.5B |       |       |
> |:-------:|:------------:|:-----:|:-----:|:-------------:|:-----:|:-----:|
> |   SFT   | $\mathcal{D}_{\textrm{SFT}}^-$ | $\mathcal{D}_{\textrm{SFT}}^+$ | $\Delta$ | $\mathcal{D}_{\textrm{SFT}}^-$ |  $\mathcal{D}_{\textrm{SFT}}^+$ | $\Delta$ |
> |  Pass@1 |     39.98    | 47.35 |  7.36 |     49.95     | 55.57 |  5.62 |
> | Pass@10 |     47.92    | 56.19 |  8.27 |     54.99     | 68.60 | 13.61 |
> |    RL   |       $\mathcal{R}_s$      |   $\mathcal{R}_{\zeta}$  | $\Delta$ |   $\mathcal{R}_s$    |   $\mathcal{R}_{\zeta}$  | $\Delta$ |
> |  Pass@1 |     33.36    | 35.17 |  1.81 |     43.92     | 44.47 |  0.55 |
> | Pass@10 |     51.22    | 56.69 |  5.47 |     63.80     | 65.91 |  2.11 |
>
> __Q2. Solution divergence has the most significant positive impact in the Pass@10 evaluation setting (vs. Pass@1 where there is much less of an impact). I am wondering whether this setting obviously benefits from more diverse solutions, and how significance the overall finding therefore is.__
>
> A2. Thank you for raising this question. We believe the benefit of solution divergence is more logically aligned with improvements in Pass@10 than in Pass@1. Under the original correctness-only reward, the model is optimized primarily for Pass@1: it can obtain high rewards even if it repeatedly produces the same correct solution. This type of optimization can cause the model to collapse onto only a few solution paths, making it more prone to generating similar failure cases when encountering more challenging problems.
>
> In contrast, introducing a divergence-based reward encourages the model not only to achieve correctness on the first try but also to explore a broader set of viable solutions during generation. This encourages diversity in reasoning patterns and problem-solving approaches. As a result, when facing difficult queries, the model is more likely to produce multiple distinct solution attempts, thereby naturally improving Pass@10 performance.

---

> ### Author Response · Authors · 2025-11-27
>
> __Q3. Further, as a control study, how would varying solution divergence of the training data by simply varying temperature, rather than based on the proposed method, compare?__
>
> A3. Thank you for the helpful suggestion. To provide a more comprehensive comparison, we conducted additional experiments by tuning the model’s temperature under the standard reward function and comparing the results with those from models trained using the divergence-introduced reward.
> From the averaged Pass@1 and Pass@10 results, we observe that increasing the temperature can indeed improve Pass@10, but this improvement comes at the cost of reduced Pass@1 performance. In contrast, models trained with the divergence-based reward achieve a more balanced outcome across both metrics.
>
> | Math-500    |                   |       |       |       |                    |       |       |       |
> |-------------|-------------------|-------|-------|-------|--------------------|-------|-------|-------|
> | Model       | Llama-3.2-1B|       |       |       | Qwen2.5-1.5B |       |       |       |
> | Reward      | $\mathcal{R}_s$ |    $\mathcal{R}_s$   |     $\mathcal{R}_s$  | $\mathcal{R}_{\zeta}$ | $\mathcal{R}_s$  |  $\mathcal{R}_s$    |   $\mathcal{R}_s$    | $\mathcal{R}_{\zeta}$ |
> | Temperature | 0.6               | 0.9   | 1.2   | 0.6   | 0.7                | 0.9   | 1.2   | 0.7   |
> | Pass@1      | 24.52             | 23.31 | 21.34 | 25.52 | 35.48              | 35.69 | 35.28 | 36.04 |
> | Pass@10     | 41.60             | 43.30 | 43.60 | 41.20 | 54.20              | 56.60 | 59.00 | 58.80 |
> | Average     | 33.06             | 33.31 | 32.47 | 33.36 | 44.84              | 46.15 | 47.14 | 47.42 |
> | __MBPP+__       |                   |       |       |       |                    |       |       |       |
> | Pass@1      | 33.40             | 33.94 | 32.00 | 33.40 | 48.10              | 44.23 | 27.02 | 47.70 |
> | Pass@10     | 44.00             | 42.92 | 49.37 | 50.00 | 59.00              | 64.38 | 64.38 | 63.00 |
> | Average     | 38.70             | 38.43 | 40.69 | 41.70 | 53.55              | 54.30 | 45.70 | 55.35 |
>
> __Q4. The text in Figure 2 is extremely hard to read. I don’t know how one would fix that, but perhaps not labelling every single model and showing just a few ones with larger font-size would work.__
>
> A4. We appreciate your great suggestion. We will only display a few representative models on the main page and one including all models in the appendix. Wish this change could be helpful for user to read the figure.
>
> __Q5. What is N in the experiment shown in Figure 2?__
>
> A5. We appreciate your question. As what we states in Line 164, we use 50 questions to calculate the solution divergence value for each model. Thus, here the N=50.

---

> ### Author Response · Authors · 2025-11-27
>
> __Q6. One could imagine many ways to incorporate solution divergence into RL rewards, and the formulation in Section 4.2 seems like one somewhat arbitrary way to do so. It would be nice to understand how this particular formulation (over any potential other ways) is motivated.__
>
> A6. Thank you for raising this question. A straightforward way to understand the RL reward design is to view the divergence reward as a reweighting mechanism. Compared with the standard +1 / –1 correctness reward, the divergence reward replaces the +1 with the average distance to other solutions. Intuitively, if the model repeatedly generates the same form of solution, these similar outputs will receive low divergence rewards because they contribute little novelty. However, if the model accidentally produces a new form of solution, the divergence reward assigns a higher value to it, encouraging the model to recognize and explore this new pattern rather than being dominated by the commonly generated ones. In this way, the divergence-based reward helps the model become more aware of diverse solution paths, ultimately improving the overall solution divergence.
>
> __Q7. It is mentioned that self-consistency (Wang et al. 2022) can be leveraged. Was it used, and if so, how?__
>
> A7. We mention the concept of self-consistency primarily to contextualize the role of the Pass@10 metric in existing studies. In our experiments, however, we do not apply this technique.
>
> __Q8. In Tables 2 and 3, data with high solution divergence is compared to data with low solution divergence. Shouldn’t there be a comparison to data with “typical” solution divergence, which would be simply a sample drawn from the model.__
>
> A8. Thank you for raising this concern. We agree that it would be more natural to adjust the solution divergence based on each model’s current performance. However, because different models exhibit different inherent divergence levels, it becomes difficult to prepare model-specific datasets in a consistent and reproducible manner. For this reason, we opted to prepare a unified dataset and conduct all experiments based on the same setup. We will explore model-adaptive divergence control in future work.
>
> __Q9. There seems to be a typo in the sentence beginning in line 430.__
>
> A9. We appreciate your suggestion. We have revised the sentence to: “Finally, because Maze is a new problem-solving task introduced in this study, using it for case analyses…”

---

### Official Review · Reviewer_zYtu · 2025-11-01

**Soundness:** 4
**Presentation:** 3
**Contribution:** 3
**Rating:** 6
**Confidence:** 3

**Summary:**

This paper introduces and investigates the concept of solution divergence — the degree to which large language models (LLMs) generate distinct solutions to the same problem — as a predictor and enhancer of problem-solving performance. The authors show that higher divergence correlates positively with better accuracy across mathematics, programming, and logical reasoning tasks.
They formalize solution divergence using pairwise normalized edit distances among generated solutions and compute global and local metrics via the Laplacian spectrum of a relational similarity graph. Empirical studies across multiple open- and closed-source LLMs demonstrate a strong correlation (R² up to 0.94) between global divergence and problem-solving success

**Strengths:**

1. The idea of quantifying and leveraging solution divergence as a performance signal for LLMs is novel.
2. The empirical correlation between divergence and performance is rigorously demonstrated across multiple datasets and 17 LLMs.
3. The theoretical foundation is clear: the Laplacian-based divergence metric ties to spectral clustering theory, providing a principled way to measure diversity in non-binary solution spaces.

**Weaknesses:**

1. Metric simplicity and linguistic bias:
i.  The divergence metric relies on normalized string edit distance, which can be brittle and insensitive to semantically distinct but lexically similar outputs (especially in math and code domains).
ii.  The authors acknowledge this but still treat it as the core metric; future iterations could integrate semantic embeddings/

2. Improvements in Pass@1 are modest (0.5–1% average in SFT, ~0.3% in RL) despite theoretical appeal, though Pass@10 gains are stronger (≈3–6%). Statistical significance results would be useful.

**Questions:**

check weakness above.

---

> ### Author Response · Authors · 2025-11-27
>
> We greatly appreciate your review and find your suggestion to provide additional evidence for demonstrating the robustness of our conclusions to be highly valuable. Below, we offer detailed responses to each of your points, and we hope these clarifications help address your concerns regarding this work.
>
> __Q1. Metric simplicity and linguistic bias. The authors acknowledge this but still treat it as the core metric; future iterations could integrate semantic embeddings.__
>
> A1. Thanks for providing the actionable suggestion. Following your guidance, we conduct the experiment with the “text-embedding-3-small” model from the OpenAI API to encode the solutions into embedding vectors and compute pairwise solution divergence using cosine similarity. We incorporated this embedding-based divergence into the RL training phase and evaluated its effect on two models: Llama-3.2-1B and Qwen2.5-1.5B.
>
> As shown in the result table, the embedding-based divergence consistently improves performance on the math-500 dataset, whereas the improvements on MBPP+ are less pronounced. We hypothesize that this difference arises from the mismatch between natural language and programming language: step-by-step math solutions resemble human-written text, making them more compatible with the capabilities of the “text-embedding-3-small” model, whereas program code is less aligned with its embedding space.
>
> |   Model  |         | Llama-3.2-1B |       |       | Qwen2.5-1.5B |       |       |
> |:--------:|:-------:|:------------:|:-----:|:-----:|:-------------:|:-----:|:-----:|
> |  Dataset |  Metric |     Edit     | Embed | $\Delta$ |      Edit     | Embed | $\Delta$ |
> | Math-500 |  Pass@1 |     25.52    | 25.63 |  0.11 |     36.04     | 37.45 |  1.41 |
> |          | Pass@10 |     41.20    | 42.28 |  1.08 |     58.80     | 60.47 |  1.67 |
> |   MBPP+  |  Pass@1 |     33.40    | 33.53 |  0.13 |     47.70     | 49.46 |  1.76 |
> |          | Pass@10 |     50.00    | 47.20 | -2.80 |     63.00     | 58.70 | -4.30 |
>
> __Q2. Improvements in Pass@1 are modest (0.5–1% average in SFT, ~0.3% in RL) despite theoretical appeal, though Pass@10 gains are stronger (≈3–6%). Statistical significance results would be useful.__
>
> A2. We appreciate your suggestion to include statistical significance in our results. Following your guidance, we performed five independent inference runs using the trained models (Llama-3.2-1B and Qwen-2.5-1.5B) on both the Math-500 and MBPP+ datasets. Below, we report the averaged results along with their statistical significance, where $\Delta$ marked with $*$ represents statistical significance with the student t-test. From these results, we observe that the differences in Pass@1 for the 0.5%–1% divergence settings are generally significant, and similarly, the differences in Pass@10 between the 3%–6% settings are also statistically significant.
>
>
> |   Model  |         | Llama-3.2-1B |            |       | Qwen-2.5-1.5B |            |       |
> |:--------:|:-------:|:------------:|:----------:|:-----:|:-------------:|:----------:|:-----:|
> |  Dataset |  Metric |     $\mathcal{R}_s$     |    $\mathcal{R}_{\zeta}$    | $\Delta$ |      $\mathcal{R}_s$     |    $\mathcal{R}_{\zeta}$    | $\Delta$ |
> | Math-500 |  Pass@1 | 24.42±0.13   | 25.65±0.21 | 1.23* | 35.62±0.15    | 36.55±0.33 | 0.82* |
> |          | Pass@10 | 41.84±0.9    | 41.45±1.28 | -0.39 | 53.90±1.66    | 57.65±0.99 | 3.75* |
> |   MBPP+  |  Pass@1 | 33.35±0.82   | 33.45±0.47 | 0.10 | 47.55±0.71    | 48.25±0.53 | 0.70* |
> |          | Pass@10 | 44.50±0.48   | 49.52±2.28 | 5.02* | 58.50±1.23    | 62.84±0.90 | 4.34* |

---

### Official Review · Reviewer_CKU3 · 2025-11-03

**Soundness:** 2
**Presentation:** 2
**Contribution:** 2
**Rating:** 4
**Confidence:** 2

**Summary:**

This paper positions solution divergence as a simple but powerful new method for understanding and improving LLM reasoning, which provides a bridge between cognitive diversity in human problem solving and data diversity in machine learning.

The authors hypothesize that models exhibiting greater diversity in their valid reasoning paths tend to have stronger general problem-solving capabilities. They quantify solution divergence using a graph-based metric derived from normalized edit distances among solutions, computed via spectral properties of the Laplacian matrix. In experiment, the authors show a strong positive correlation  between solution divergence and success rate (Pass@1) across multiple LLMs.

**Strengths:**

1. This paper proposes a conceptual framing. The propose concept is original and intuitive.
2. Results are consistent across multiple reasoning domains (math, logic, programming) and different model families (GPT-4o, Claude, Gemini, Llama-3, Qwen).
3.

**Weaknesses:**

1. The paper establishes a strong correlation between solution divergence and model accuracy, but causal mechanisms remain unclear. It remains unknown whether higher divergence causes better reasoning, or whether better models naturally produce both diverse and correct solutions.
2. Although the experiments span math, logic, and programming, these tasks are still structured and verifiable.
3. The paper focuses on how diversity improves performance, but doesn’t analyze potential downsides. For example, could encouraging divergence lead to less stable or less factual reasoning?

**Questions:**

1. Is there a case where some diversities are bad and lead to incoherent or invalid reasoning?

---

> ### Author Response · Authors · 2025-11-27
>
> We appreciate your thoughtful feedback and the valuable questions raised in your review. Below, we provide detailed responses to each point, and we hope these clarifications help you evaluate our paper with a more comprehensive understanding of our intentions, methods, and findings.
>
> __Q1. It remains unknown whether higher divergence causes better reasoning or whether better models naturally produce both diverse and correct solutions.__
>
> A1. By conducting experiments across three representative problem-solving datasets and consistently observing the positive influence of incorporating solution divergence, we aim to demonstrate its empirical effectiveness rather than claim strict causation. Additionally, solution divergence can be interpreted as a form of regularization during training, especially in the RL phase, where it helps prevent the model from prematurely converging to a single correct solution. Instead, it encourages the model to better understand the problem and learn a broader distribution of valid solution pathways.
>
> __Q2. Although the experiments span math, logic, and programming, these tasks are still structured and verifiable.__
>
> A2. As we mentioned in line 142, we choose those datasets that are the representative ones for problems solving tasks and the verifiable datasets make the experiment easier to reproduce and follow by the others. We will put the exploring on more forms of tasks in our future studies.
>
> __Q3. The paper focuses on how diversity improves performance, but doesn’t analyze potential downsides. For example, could encouraging divergence lead to less stable or less factual reasoning?__
>
> A3. Thank you for the thoughtful comment. We indeed observed several interesting behaviors during training. For example, in the RL phase for math problems, the model occasionally attempted to “hack” the correctness verification function by adding unnecessary decorative content—such as repeating the final answer multiple times—in order to obtain higher solution-divergence rewards. To suppress such behavior, we incorporated a format reward to check for redundant steps or irrelevant content, as well as a length-penalty reward to discourage overly long responses.
> Another noteworthy finding involves the Qwen model. Because it is trained by an international research team, we observed that during RL training it would sometimes generate multilingual responses (e.g., mixed Chinese and English) within a single solution.

---

### Official Review · Reviewer_98hL · 2025-11-04

**Soundness:** 2
**Presentation:** 3
**Contribution:** 3
**Rating:** 4
**Confidence:** 3

**Summary:**

This paper introduces and investigates the concept of "solution divergence," which refers to the existence of different yet equally valid solutions to a single problem. The authors' preliminary study reveals a strong positive correlation between an LLM's ability to generate diverse solutions and its overall problem-solving performance across tasks in mathematics, programming, and logical reasoning.

Building on this finding, the paper proposes two methods to leverage solution divergence to improve LLM training:

- Dataset Divergence Metric (for SFT): A method for Supervised Fine-Tuning (SFT) where training data is selected based on its inherent solution diversity. Models trained on datasets with higher divergence showed improved performance.
- Solution Divergence Fused Reward (for RL): A novel reward function for Reinforcement Learning (RL) that encourages the model to generate solutions that are not only correct but also different from other solutions. This method also led to significant performance gains, particularly in finding a correct answer within multiple attempts (Pass@10).

In essence, the research demonstrates that solution divergence is a simple but powerful metric that can be effectively integrated into both SFT and RL pipelines to enhance the problem-solving capabilities of large language models.

**Strengths:**

- This paper introduces the novel and underexplored concept of "solution divergence" as a key factor in LLM problem-solving performance. And it provides strong empirical evidence demonstrating a positive correlation between a model's ability to generate diverse solutions and its success rate across different domains.
- It proposes two practical and actionable methods for integrating solution divergence into standard LLM training pipelines: one for SFT and one for RL.
- The research is grounded in a comprehensive experimental setup, utilizing three distinct datasets (Math-500, MBPP+, and the newly created Maze), a wide array of 17 different LLMs, and robust evaluation.
- The authors formalize the abstract concept of divergence into a computable metric using graph theory and string edit distance, making the approach replicable.

**Weaknesses:**

- The core metric of "solution divergence" relies on a superficial and potentially misleading proxy. The entire paper's foundation rests on measuring divergence using normalized string edit distance. This is a crude, syntactic measure that is blind to semantic, logical, or structural equivalence. Two solutions could be algorithmically identical but written with different variable names or sentence structures, leading to a high, artificial divergence score. Conversely, two genuinely different logical approaches could be expressed with similar vocabulary, yielding an erroneously low score. This fundamental weakness calls into question the validity of the paper's core findings.

- The paper risks conflating correlation with causation. The preliminary study only establishes a correlation: better models tend to produce more diverse solutions. The paper then leaps to the conclusion that forcing more divergence into training causes better performance. An equally plausible hypothesis is that stronger, more capable models naturally develop a wider range of strategies as a byproduct of their advanced reasoning abilities, making divergence a symptom of competence, not its cause.

- The heavy reliance on the novel "Maze" dataset for key ablation studies is a significant weakness. The Maze task, created for this study, is an unvetted, toy problem. It is highly structured and path-based, making it uniquely and perhaps artificially sensitive to a syntactic diversity metric. The most compelling results and ablation studies are performed on this dataset, but there is no guarantee these findings would hold for more complex, established benchmarks. The minimal impact on the MBPP+ dataset could be an indicator that the effect is not as general as claimed.

- The method for generating SFT data introduces a major confounding variable. The authors use powerful, state-of-the-art LLMs (GPT-4o, Claude-3.5) to generate the "diverse" solutions for their training sets. The subsequent training of smaller models on this data may not be learning "divergence" as a principle, but rather is simply a form of knowledge distillation from a superior teacher model. The performance gains could be attributed to the quality of the distilled solutions, not the diversity metric used to select them.

- The practical application and scalability of the proposed SFT method are questionable. The process of creating the high-divergence dataset involves generating a large pool of solutions, enumerating all possible subsets (e.g., all 4-solution subsets), and calculating divergence for each one. This is a combinatorial explosion that is computationally infeasible for any reasonably sized pool of candidate solutions or larger subset selections.

**Questions:**

See weaknesses.

---

> ### Author Response · Authors · 2025-11-27
>
> __Q1. The entire paper's foundation rests on measuring divergence using normalized string edit distance. This is a crude, syntactic measure that is blind to semantic, logical, or structural equivalence.__
> A1. We appreciate your comment. As noted in lines 125–126, we acknowledge that our current divergence estimation using normalized edit distance is not perfect and comes with certain limitations. We chose this metric primarily for its computational simplicity and broad applicability across different tasks.
>
> To examine how alternative divergence metrics might influence our findings, we conducted an additional experiment using a more semantic-based approach. Specifically, we employed the “text-embedding-3-small” model from the OpenAI API to encode the solutions into embedding vectors and computed pairwise solution divergence using cosine similarity. We incorporated this embedding-based divergence into the RL training phase and evaluated its effect on two models: Llama-3.2-1B and Qwen-2.5-1.5B.
>
> As shown in the result table, the embedding-based divergence consistently improves performance on the math-500 dataset, whereas the improvements on MBPP+ are less pronounced. We hypothesize that this difference arises from the mismatch between natural language and programming language: step-by-step math solutions resemble human-written text, making them more compatible with the capabilities of the “text-embedding-3-small” model, whereas program code is less aligned with its embedding space.
>
> |   Model  |         | Llama-3.2-1B |       |       | Qwen-2.5-1.5B |       |       |
> |:--------:|:-------:|:------------:|:-----:|:-----:|:-------------:|:-----:|:-----:|
> |  Dataset |  Metric |     Edit     | Embed | $\Delta$ |      Edit     | Embed | $\Delta$ |
> | Math-500 |  Pass@1 |     25.52    | 25.63 |  0.11 |     36.04     | 37.45 |  1.41 |
> |          | Pass@10 |     41.20    | 42.28 |  1.08 |     58.80     | 60.47 |  1.67 |
> |   MBPP+  |  Pass@1 |     33.40    | 33.53 |  0.13 |     47.70     | 49.46 |  1.76 |
> |          | Pass@10 |     50.00    | 47.20 | -2.80 |     63.00     | 58.70 | -4.30 |
>
> __Q2. The paper risks conflating correlation with causation. The preliminary study only establishes a correlation: better models tend to produce more diverse solutions. The paper then leaps to the conclusion that forcing more divergence into training causes better performance.__
>
> A2. We agree that our results do not imply a direct causal relationship between improved problem-solving capability and increased solution divergence. This is precisely why we designed the study as an empirical investigation. By conducting experiments across three representative problem-solving datasets and consistently observing the positive influence of incorporating solution divergence, we aim to demonstrate its empirical effectiveness rather than claim strict causation. Additionally, solution divergence can be interpreted as a form of regularization during training, especially in the RL phase, where it helps prevent the model from prematurely converging to a single correct solution. Instead, it encourages the model to better understand the problem and learn a broader distribution of valid solution pathways.

---

> ### Author Response · Authors · 2025-11-27
>
> __Q3. The heavy reliance on the novel "Maze" dataset for key ablation studies is a significant weakness. The minimal impact on the MBPP+ dataset could be an indicator that the effect is not as general as claimed.__
>
> A3. As we mentioned in line 377, the limitation on dataset size of the MBPP may influence the significance of its results. To solve that concern, we rerun the experiment 2 times more using the random splits with different seed and with the model Llama-3.2-1B and Qwen2.5-1.5B and report the average performance. From the table, we can find the average performance indicate the better consistency with the other two datasets.
>
> |  Metric | Llama-3.2-1B |       |       | Qwen2.5-1.5B |       |       |
> |:-------:|:------------:|:-----:|:-----:|:-------------:|:-----:|:-----:|
> |   SFT   | $\mathcal{D}_{\textrm{SFT}}^-$ | $\mathcal{D}_{\textrm{SFT}}^+$ | $\Delta$ | $\mathcal{D}_{\textrm{SFT}}^-$ |  $\mathcal{D}_{\textrm{SFT}}^+$ | $\Delta$ |
> |  Pass@1 |     39.98    | 47.35 |  7.36 |     49.95     | 55.57 |  5.62 |
> | Pass@10 |     47.92    | 56.19 |  8.27 |     54.99     | 68.60 | 13.61 |
> |    RL   |       $\mathcal{R}_s$      |   $\mathcal{R}_{\zeta}$  | $\Delta$ |   $\mathcal{R}_s$    |   $\mathcal{R}_{\zeta}$  | $\Delta$ |
> |  Pass@1 |     33.36    | 35.17 |  1.81 |     43.92     | 44.47 |  0.55 |
> | Pass@10 |     51.22    | 56.69 |  5.47 |     63.80     | 65.91 |  2.11 |
>
>
> __Q4. The subsequent training of smaller models on this data may not be learning "divergence" as a principle, but rather is simply a form of knowledge distillation from a superior teacher model. The performance gains could be attributed to the quality of the distilled solutions, not the diversity metric used to select them.__
>
> A4. We choose to use this way for the SFT is because it is the most straightforward way to make the solution divergence as a controllable variable for our experiment. In addition, during the data preparation, we have verified all the solutions are correct and we there is no error in the steps. So we believe the data quality for the two datasets should at the same level.
>
>
> __Q5. The practical application and scalability of the proposed SFT method are questionable. The process of creating the high-divergence dataset involves generating a large pool of solutions, enumerating all possible subsets (e.g., all 4-solution subsets), and calculating divergence for each one.__
>
> A5. In fact, we use the combination during the preparation for the SFT dataset is designed for have the controls over the divergence metric for the later experiments. In practice, there is no need to exhaust all the available solutions, but simply using all the available correct solutions during preparation.

---

### Author Response · Authors · 2025-12-02
**Author Team Rebuttal Summary**

We would like to express our sincere appreciation to all four reviewers and the ACs for your comments, suggestions, and support on this paper, especially given the difficulty of this review period. By carefully studying and addressing your feedback, we believe the revised manuscript has substantially improved in clarity, rigor, and overall quality.

Below, we summarize the major comments raised across the four reviews, along with our general responses and corresponding revision actions. Detailed, point-by-point replies are provided in the individual reviewer response sections.

Overall, reviewers provided positive feedback on the presentation quality and remained cautiously positive about both the contribution and methodological soundness. All four reviewers acknowledged the novelty of our work in examining the relationship between LLM solution divergence and problem-solving performance. They also appreciated the comprehensive experiments across three representative problem-solving datasets and recognized the effectiveness of our proposed divergence-aware training frameworks. At the same time, reviewers raised several shared concerns, which we address below.

(1) Concern on divergence computation via edit distance (Reviewers 98hL, zYtu).
In response, we conducted an additional experiment (Lines 961–969) using the high-performing semantic embedding model text-embedding-3-small to compute solution divergence during the RL stage and compared it against edit-distance divergence. The results confirm our claim in Section 3.1: although edit distance is not a perfect measure, it is effective for most general-purpose cases, while the embedding-based method, despite its strengths, can be less reliable for solutions with highly structured formats.

(2) Concerns regarding stability and statistical significance (Reviewers 98hL, zYtu, i9wi).
To reduce the influence of randomness, we added repeated-run experiments (Lines 950–957) and conducted Student’s t-tests to evaluate statistical significance. We additionally performed a new random split of MBPP+ and reported averaged results to demonstrate more consistent alignment across all datasets.

(3) Clarification of the mechanism linking solution divergence to performance gains (Reviewers 98hL, CKU3).
We expanded the explanation in Lines 297–301. In particular, during RL, divergence acts as a reweighting factor that balances the learning signal between dominant solution clusters and correct but underexplored creative solutions. Encouraging this balance enables the model to learn from a wider range of behaviors.

(4) Suggestion to compare against temperature tuning during inference (Reviewer i9wi).
We appreciate this suggestion and added results for all three datasets (Lines 483–527), showing that our method consistently outperforms simple temperature adjustments.

(5) Requests for clarification on experimental settings and edge-case outcomes (Reviewers CKU3, i9wi).
We have added the requested details both in the rebuttal and in the revised manuscript.

In total, our revision adds embedding-based divergence experiments, repeated-run significance analyses, temperature comparisons, and expanded methodological clarifications. We believe these additions thoroughly address the reviewers’ concerns. Although it is unfortunate that we cannot engage in further discussion during this challenging period, we hope our responses satisfactorily resolve all remaining questions.

We sincerely thank all reviewers and ACs again for your time, effort, and thoughtful feedback.

The Authors

---

### Note · Authors · 2026-01-06

I have read and agree with the venue's withdrawal policy on behalf of myself and my co-authors.